# Ep400 deficiency in Schwann cells causes persistent expression of early developmental regulators and peripheral neuropathy

Franziska Fröb [1], Elisabeth Sock [1], Ernst.R. Tamm[2], Anna-Lena Saur[1], Simone Hillgärtner[1], Trevor J. Williams[3], Toshihiro Fujii[4], Rikiro Fukunaga [4] & Michael Wegner [1]

Schwann cells ensure efficient nerve impulse conduction in the peripheral nervous system. Their development is accompanied by defined chromatin changes, including variant histone deposition and redistribution. To study the importance of variant histones for Schwann cell development, we altered their genomic distribution by conditionally deleting Ep400, the central subunit of the Tip60/Ep400 complex. Ep400 absence causes peripheral neuropathy in mice, characterized by terminal differentiation defects in myelinating and non-myelinating Schwann cells and immune cell activation. Variant histone H2A.Z is differently distributed throughout the genome and remains at promoters of *Tfap2a*, *Pax3* and other transcriptional regulator genes with transient function at earlier developmental stages. Tfap2a deletion in Ep400-deficient Schwann cells causes a partial rescue arguing that continued expression of early regulators mediates the phenotypic defects. Our results show that proper genomic distribution of variant histones is essential for Schwann cell differentiation, and assign importance to Ep400-containing chromatin remodelers in the process.

[1] Institut für Biochemie, Emil-Fischer-Zentrum, Friedrich-Alexander-Universität Erlangen-Nürnberg, Fahrstrasse 17, 91054 Erlangen, Germany. [2] Institut für Humananatomie und Embryologie, Universität Regensburg, Universitätsstrasse 31, 93053 Regensburg, Germany. [3] Departments of Craniofacial Biology and Cell and Developmental Biology, UC Denver, Anschutz Medical Campus, 12801 East 17th Avenue, Aurora, CO 80045, USA. [4] Department of Biochemistry, Osaka University of Pharmaceutical Sciences, 4-20-1 Nasahara, Takatsuki, Osaka 569-1094, Japan. Correspondence and requests for materials should be addressed to M.W. (email: michael.wegner@fau.de)

In the vertebrate peripheral nervous system (PNS), efficient conduction of nerve impulses requires Schwann cells (SCs) and their myelin. SC development from neural crest cells is coordinated by sets of transcriptional regulators with defined windows of expression[1]. These include Tfap2a and Pax3 in the early neural crest and SC precursor stages, Sox2 in the immature, and Oct6 in the pro-myelinating stage as well as Krox20 as a central regulator of the final differentiation and myelination process[2]. Additionally, Sox10 is present and required throughout SC development from specification to maintenance of the mature myelinating state[3].

SC development also depends on timely changes in chromatin structure[4]. In addition to histone modifications and alterations in nucleosome positioning, such chromatin changes usually involve region-specific exchanges of standard histones against variant histones that are performed by chromatin remodeling complexes of the SWR and INO80 type[5]. H2A.Z is such a variant histone that can replace H2A. H2A.Z has limited sequence identity to H2A so that its incorporation changes nucleosome properties and influences transcription. In cellular systems, activating and repressing effects on transcription initiation and elongation have been described for H2A.Z[6,7]. Very few studies exist on the role of variant histones in development[8,9].

To address the importance of proper variant histone deposition for SCs and peripheral myelination, we deleted Ep400 at different times of SC development. Ep400 is the central ATP-hydrolyzing subunit of the prototypical SWR-type Tip60/Ep400 complex. This chromatin remodeler exchanges H2A.Z against H2A in nucleosomes[10,11]. It has also been reported to simultaneously replace the standard H3.1 by H3.3 in regulatory regions[12]. By deleting Ep400, the complex loses its chromatin remodeling activity, leading to defects in cell renewal, pluripotency, cell-cycle progression, cell survival, DNA repair, and apoptosis in cellular systems[13–15].

Here we show that SC-specific deletion of Ep400 in the mouse causes defects in late stages of SC development and peripheral myelination. Our results argue that altered genomic H2A.Z distribution leads to a failure to shut off early developmental regulators whose continued presence in differentiating SCs interferes with the maturation and myelination process.

## Results

### Ep400 expression in SCs.
We generated antibodies against Ep400 to investigate its occurrence in SCs during development and PNS myelination. Starting at embryonic day (E) 12.5, Ep400 immunoreactivity was detected along spinal nerves in SCs marked by Sox10 expression. Ep400 remained present in Sox10-positive cells not only during prenatal development until E18.5 (Supplementary Fig. 1a–d) but was also found in Sox10-positive cells of the sciatic nerve at P9, P21, and at 2 months of age (Supplementary Fig. 1e–g). During this time, Sox10-positive cells of the SC lineage progress from SC precursor via immature, pro-myelinating, and myelinating stages into a fully mature SC. The continuous detection argues that Ep400 is present at all times of SC development and in the adult.

For confirmation, co-localization of Ep400 with stage-specific SC markers was analyzed by immunofluorescence. Ep400 was indeed found in Sox2-positive immature SCs, Oct6-positive pro-myelinating SCs, and Krox20-positive myelinating SCs (Supplementary Fig. 1h–j).

Other cell types in the peripheral nerve also expressed Ep400 (Supplementary Fig. 1k). These included Iba1-positive macrophages, CD3-positive T lymphocytes, α-smooth muscle actin-positive perivascular smooth muscle cells, Pecam-positive endothelial cells, Desmin-positive pericytes, and fibronectin-positive fibroblasts.

### Peripheral neuropathy in mice with SC-specific Ep400 deletion.
To prevent Ep400 expression in SCs, we first combined the Ep400[fl] allele[14] with a Sox10::Cre BAC transgene[16]. This allowed efficient Ep400 deletion during early neural crest development (Supplementary Fig. 2a). At E12.5, the resulting Ep400ΔNC mice still possessed Sox10- and Fabp7-positive SC precursors along spinal nerves (Supplementary Fig. 2b–e). This argues that Ep400 is not essentially required for SC specification. The Sox10::Cre transgene deletes widely throughout the neural crest. As a consequence Ep400ΔNC mice exhibited neural crest-related abnormalities such as cleft lip, cleft palate, and other craniofacial malformations and died at birth (Supplementary Fig. 2f).

To investigate SC development postnatally, we combined the Ep400[fl] allele and a Dhh::Cre transgene[17]. In the resulting Ep400ΔPNS mice, Ep400 was deleted specifically in SCs at the late precursor or early immature SC stage[18]. By the time of birth, >90% of all SCs did not contain detectable levels of Ep400 protein (Fig. 1a–d, Supplementary Fig. 3a). Ep400ΔPNS mice were born at normal Mendelian ratios but became distinguishable from their control littermates around P14, when pups started to explore their environment. They exhibited poor motor coordination and an unsteady gait as characteristic symptoms of a peripheral neuropathy. Motor deficits persisted. At P21, Ep400ΔPNS mice had reduced grip strength, clasped their hind limbs when lifted by their tails (Fig. 1e, g), and sciatic nerves were more translucent (Fig. 1f, h). While Ep400ΔPNS mice survived well during the first 2 months of their life, their condition worsened with age (Supplementary Fig. 3b). Few mice grew older than 5 months.

Ultrastructural and histological analyses of the sciatic nerve revealed a severe hypomyelination at P21 in Ep400ΔPNS mice. Approximately 25% of large caliber axons were still unmyelinated in the mutant at times when axons were fully myelinated in the control (Fig. 1i–r, Supplementary Fig. 3c–h). The vast majority of these unmyelinated axons were in 1:1 contact with SCs (Fig. 1k, arrow), indicating normal lineage progression until arrest at the pro-myelinating stage. Those axons that were myelinated exhibited a substantially thinner myelin sheath (Fig. 1k, arrowhead, Supplementary Fig. 3g, h) as indicated by an increase in the mean g ratio (defined as ratio of inner axonal diameter to total outer fiber diameter) from approximately 0.65 in controls to 0.70 (Fig. 1i–q). The largest caliber axons appeared most affected (Supplementary Fig. 3k, l). Remak bundles as associations of SCs with groups of small caliber axons were fewer and had an altered appearance in Ep400ΔPNS mice (Fig. 1j, l, Supplementary Fig. 3w). At P21, considerable amounts of cytoplasm remained present between axons and a large fraction of axons were not completely surrounded by SC membrane. Already at this time, activated macrophages were found in the nerve (Fig. 1s). Signs of abnormal and degraded myelin were widespread (Fig. 1t). At 2 months, myelination had improved in Ep400ΔPNS mice with few unmyelinated axons remaining and g ratios only slightly higher than in control mice (Fig. 1m–v, Supplementary Fig. 3e–j). Only axons with the largest caliber still exhibited a significant increase in g ratio as compared to controls (Supplementary Fig. 3m, n). However, myelin degradation was still ongoing (Fig. 1p, w, asterisks). We also detected many myelin outfoldings or other unusual structures such as myelin around an established myelin sheath that still contained an axon or had already lost it (Fig. 1o, x). Myelin debris and myelin outfoldings in Ep400ΔPNS mice were comparably increased at both ages (Supplementary Fig. 3s–v). Remak bundles also remained fewer and structurally abnormal at 2 months (Fig. 1n, p, Supplementary Fig. 3x).

In addition, we detected severe changes in myelin sheath length. Average internodal length in teased fibers was dramatically decreased to 56 μm in Ep400ΔPNS mice as compared to 119 μm in controls (Fig. 2a). Furthermore, none of the internodes

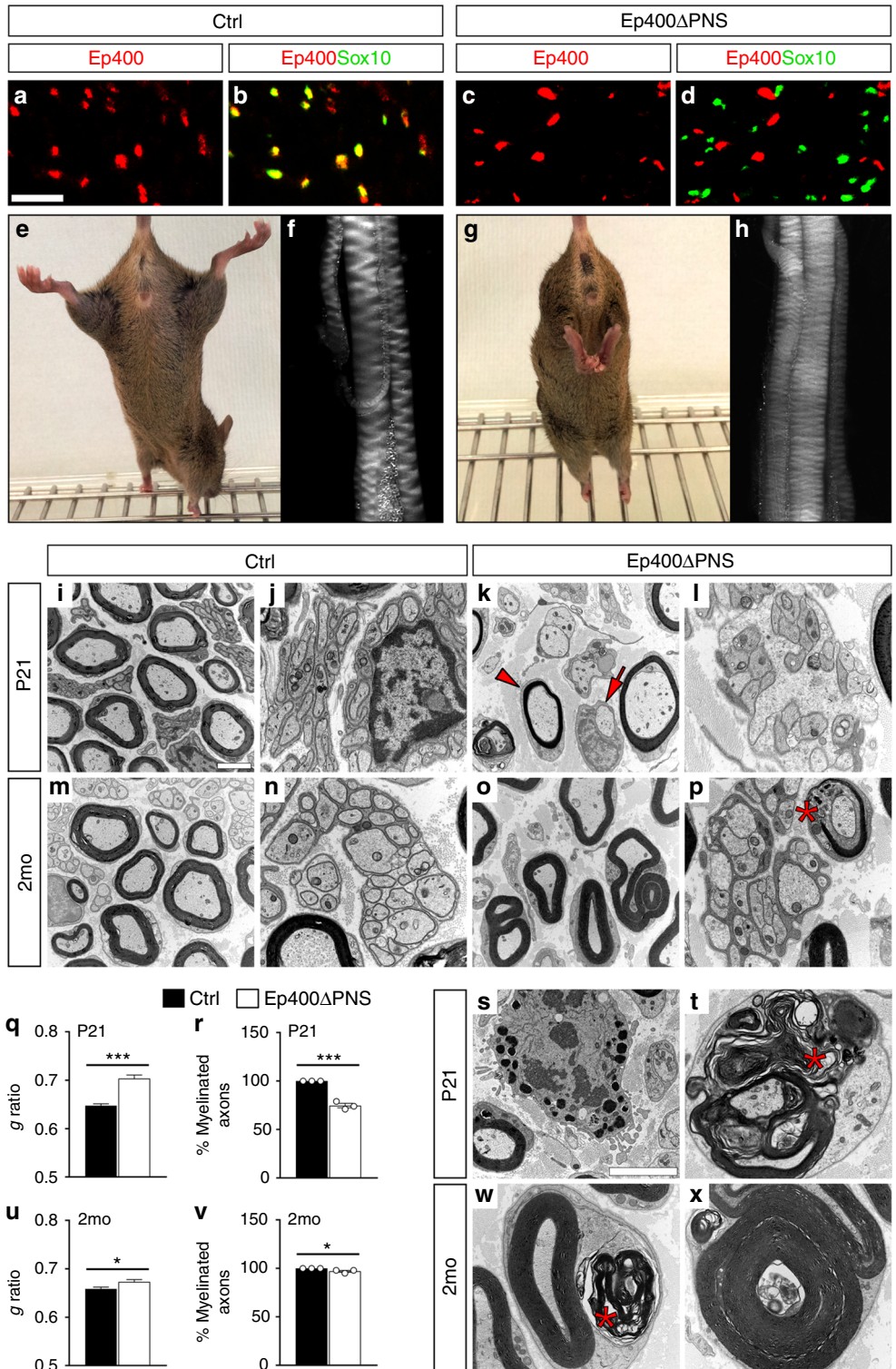

**Fig. 1** Peripheral neuropathy resulting from Ep400 deletion in Schwann cells (SCs). **a**–**d** Occurrence of Ep400 in SCs of sciatic nerves from control (**a**, **b**) and *Ep400ΔPNS* (**c**, **d**) mice at P21 as determined by co-immunofluorescence studies with antibodies against Ep400 (red) and Sox10 (green) to prove efficient SC-specific deletion. Sox10-negative cells in the nerve retained Ep400 and may represent endoneurial fibroblasts, pericytes, endothelial cells, or immune cells. Scale bar: 25 μm. **e**–**h** Hindlimb clasping phenotype (**e**, **g**) and sciatic nerve hypomyelination (**f**, **h**) in *Ep400ΔPNS* (**g**, **h**) as compared to control (**e**, **f**) mice at P21. **i**–**p**, **s**, **t**, **w**, **x** Representative electron microscopic pictures of sciatic nerve sections from control (**i**, **j**, **m**, **n**) and *Ep400ΔPNS* (**k**, **l**, **o**, **p**, **s**, **t**, **w**, **x**) mice at P21 (**i**–**l**, **s**, **t**) and 2 months (2 mo) (**m**–**p**, **w**, **x**) in overview (**i**–**p**) and at higher resolution (**s**, **t**, **w**, **x**). Magnifications depict an activated macrophage (**s**) and various myelin abnormalities (**t**, **w**, **x**). Arrow, unmyelinated axon; arrowhead, hypomyelinated axon; asterisk, myelin debris. Scale bars: 2.5 μm. **q**, **r**, **u**, **v** Determination of the mean *g* ratio (**q**, **u**) and the number of myelinated axons as percentage of total axons with a diameter ≥1 μm (**r**, **v**) in ultrathin sciatic nerve sections of control (black bars) and *Ep400ΔPNS* (white bars) mice at P21 (**q**, **r**) and 2 mo (**u**, **v**). All large caliber axons were myelinated in control mice. Statistical significance was determined by unpaired, two-tailed Student's *t* test (*$P \le 0.05$; ***$P \le 0.001$). Exact values are listed in Supplementary Tables 1 and 2 and source data are provided as a Source Data file

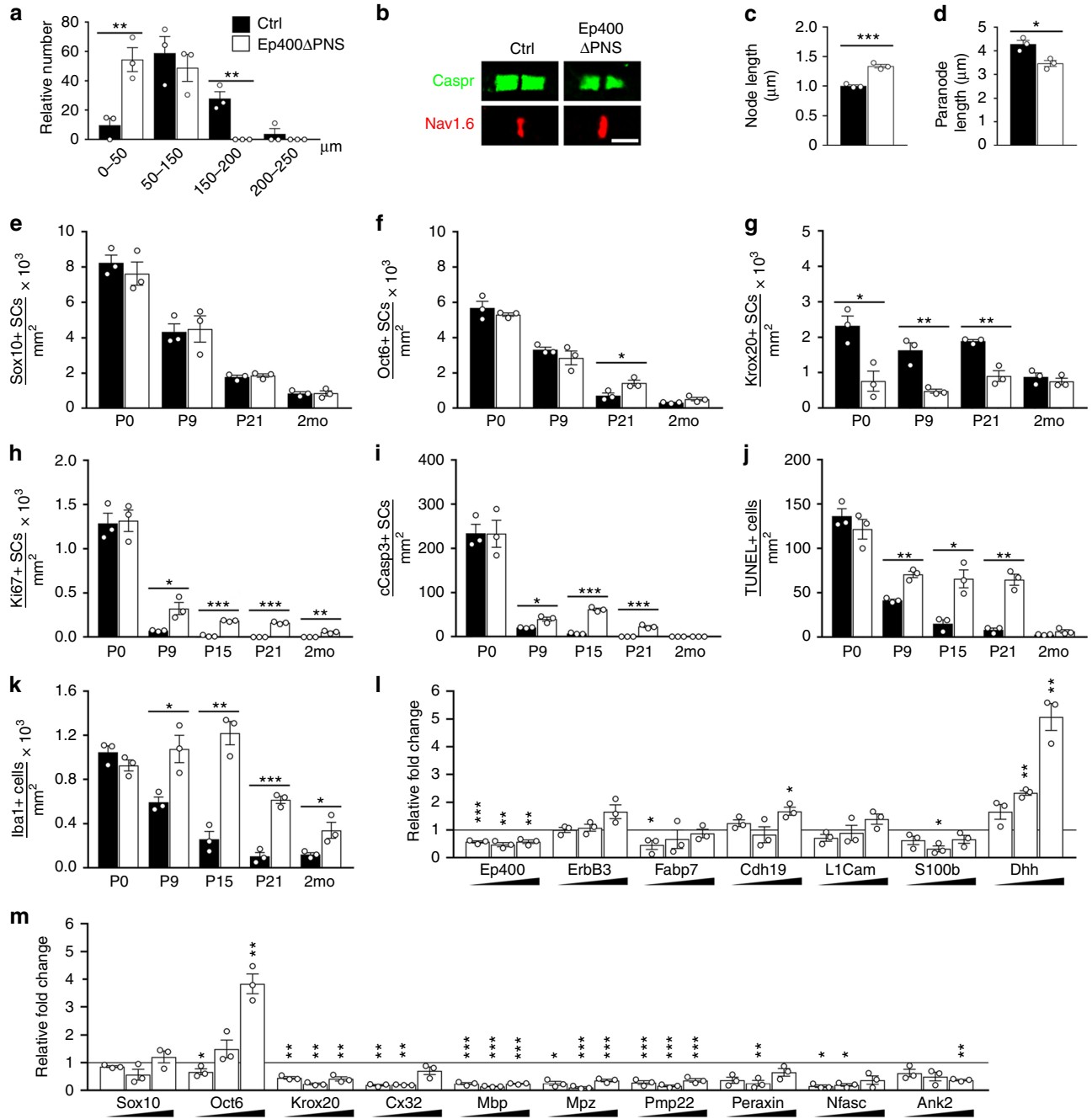

**Fig. 2** Sciatic nerve features in *Ep400ΔPNS* mice. **a** Relative distribution of internodal length in stained teased fibers from sciatic nerves of control (black bars) and *Ep400ΔPNS* (white bars) mice at P21 (*n* = 3, approximately 50 internodes measured per nerve). **b** Representative confocal images of paranode (green, for Caspr) and node (red, for Nav1.6) stainings in teased fibers obtained from sciatic nerves of control and *Ep400ΔPNS* mice at P21 by immunohistochemistry. Scale bar: 5 μm. **c, d** Length determination of node and paranode in sciatic nerves of control and *Ep400ΔPNS* mice at P21 from confocal images of teased fibers after immunohistochemistry (*n* = 3; mean values ± SEM). **e–k** Comparison of the total number of Sox10-positive Schwann cells (SCs) (**e**), Oct6-positive pro-myelinating SCs (**f**), Krox20-positive myelinating SCs (**g**), Ki67-positive proliferating SCs (**h**), cleaved Caspase 3-positive apoptotic SCs (**i**), total TUNEL-labeled cells (**j**), and Iba1-positive macrophages (**k**) in sciatic nerve sections of control and *Ep400ΔPNS* mice at P0, P9, P15, P21, and at 2 months (2mo) of age (*n* = 3; mean values ± SEM). **l, m** Determination of transcript levels for *Ep400, ErbB3, Fabp7, Cdh19, L1Cam, S100b, Dhh, Sox10, Oct6, Krox20, Connexin-32* (Cx32), *Mbp, Mpz, Pmp22, Periaxin, Neurofascin* (*Nfasc*), and *Ankyrin 2* (*Ank2*) in sciatic nerve of *Ep400ΔPNS* mice at increasing age (P9, P21, and 2mo from left to right, indicated by triangle below the bars) by quantitative reverse transcriptase PCR as compared to transcript levels in age-matched control tissue, which were arbitrarily set to 1 (marked by line) (*n* = 3; mean values ± SEM). Statistical significance was determined for a particular parameter or gene at a particular age between control and *Ep400ΔPNS* mice by two-tailed Student's *t* test (*P ≤ 0.05; **P ≤ 0.01; ***P ≤ 0.001). Exact values are listed in Supplementary Tables 3–6 and source data are provided as a Source Data file

in *Ep400ΔPNS* mice surpassed 150 μm in length, whereas this was the case for 37% of all internodes in control mice. By staining with antibodies against Nav1.6 as marker of the node of Ranvier and Caspr as paranodal marker (Fig. 2b), we detected a substantial 33% lengthening of the node (Fig. 2c) and a 19% shortening of the paranode in *Ep400ΔPNS* mice (Fig. 2d). Changes were confirmed by electron microscopy of sciatic nerves at P21 and 2 months of age (Supplementary Fig. 3o–r; average nodal length of 1.0 μm in control and 1.6 μm in *Ep400ΔPNS* mice, average paranodal length of 4.2 μm in control and 3.1 μm in *Ep400ΔPNS* mice).

Immunohistochemical staining of transverse sciatic nerve sections with antibodies against Sox10 as a pan-SC marker[19] revealed normal SC numbers along the nerve of *Ep400ΔPNS* mice from birth until 2 months of age (Fig. 2e, Supplementary Fig. 4a–h). We also failed to detect differences in Oct6 occurrence between *Ep400ΔPNS* mice and controls during the active phase of myelination in the first 2 postnatal weeks (Fig. 2f, Supplementary Fig. 4i–p). As Oct6 is a marker of pro-myelinating SCs[20,21], these results argue that SC development in *Ep400ΔPNS* mice proceeded into this stage. From P21 onwards, we observed a slight increase of Oct6-positive cells along nerves of *Ep400ΔPNS* mice (Fig. 2f, Supplementary Fig. 4k–p).

Dramatic alterations were detected for Krox20, a transcription factor induced early during myelination as the central regulator of this process in SCs[22]. The number of Krox20-positive SCs was already decreased at birth in *Ep400ΔPNS* mice and remained substantially lower than in controls until P21 (Fig. 2g, Supplementary Fig. 4q–w). Recovery to normal levels was not achieved before 2 months of age (Fig. 2g, Supplementary Fig. 4t, x). Reduced Krox20 expression should impact myelin gene expression. This was confirmed by staining with antibodies against Mbp (Supplementary Fig. 3c–i). SC development was therefore affected at the myelinating stage in *Ep400ΔPNS* mice.

SC proliferation was still normal around birth (Fig. 2h) with a similar proportion of SCs actively dividing (Supplementary Fig. 5a), residing in S phase (Supplementary Fig. 5b, c), and exiting the cell cycle as in controls (Supplementary Fig. 5d). However, total proliferation (Supplementary Fig. 5e, f) and SC-specific proliferation (Fig. 2h) were significantly higher in sciatic nerves of *Ep400ΔPNS* mice than in controls from P9 onwards. Apoptosis was simultaneously increased as measured by cleaved caspase 3 staining or terminal deoxynucleotidyl transferase-mediated dUTP-fluorescein nick end labeling (TUNEL; Fig. 2i, j, Supplementary Fig. 5g–i). Approximately half the dying cells were identified as SCs by staining for Sox10 (Fig. 2i).

Confirming the results from ultrastructural analysis (Fig. 1s), Iba1-positive macrophages were also increased in the nerves of *Ep400ΔPNS* mice from P9 onwards (Fig. 2k, Supplementary Fig. 5j). Macrophage proliferation likely contributed to the increased proliferation in the nerves of *Ep400ΔPNS* mice. The number of macrophages remained elevated even at 2 months.

We conclude from these findings that SC differentiation in *Ep400ΔPNS* mice was perturbed early during myelination. This was accompanied with a slightly increased rate of apoptosis that is compensated by a similarly increased rate of proliferation. By 2 months, the overall appearance of nerves and their myelin in *Ep400ΔPNS* mice resembled the control. However, ultrastructural abnormalities in myelin and signs of inflammation persisted. Quantitative reverse transcriptase PCR (qRT-PCR) analysis revealed that many markers of earlier phases of SC development exhibited comparable expression in sciatic nerves of control and *Ep400ΔPNS* mice at P9, P21, and 2 months (Fig. 2l). This included *ErbB3*, *Fabp7*, *Cdh19*, *L1Cam*, and *S100b*. Only few genes exhibited substantial upregulation such as *Dhh* and *Oct6* (Fig. 2l, m) arguing that gene expression was not globally

deregulated in Ep400-deficient SCs. In contrast, the expression of *Krox20* and many myelin-associated genes such as *Connexin-32*, *Mbp*, *Mpz*, *Pmp22*, *Periaxin*, and *Neurofascin* was substantially lower in *Ep400ΔPNS* mice than in controls (Fig. 2m). This was still the case at 2 months arguing that myelin gene expression remained reduced in adult *Ep400ΔPNS* mice despite the observed normalization in the number of myelinating SCs.

**Changed expression and chromatin marks in *Ep400ΔPNS* nerves.** To understand the cause of the peripheral neuropathy, we performed RNA-Seq on sciatic nerves of *Ep400ΔPNS* mice and controls at P9 as a time point that exhibited the earliest phenotypical changes and allowed reliable isolation of high-quality RNA in sufficient quantities. In principal component analysis (PCA), RNA samples from *Ep400ΔPNS* mice and controls clustered separately (Supplementary Fig. 6a). As expected for tissues that contain different cell types, of which only one was directly affected by gene deletion, effect sizes were moderate for most genes with altered expression (Supplementary Fig. 6b). Only 1056 genes (corresponding to 4.3% of all detected genes) were upregulated more than two-fold and 791 genes (corresponding to 3.2%) were downregulated more than two-fold in the nerves of *Ep400ΔPNS* mice as compared to control (Fig. 3a, Supplementary Fig. 6b).

According to gene ontology (GO) analysis, upregulated genes mainly fell into one of three categories. A large fraction was associated with immune system, immune defense, and inflammatory response (Fig. 3b). Gene set enrichment analysis (GSEA) confirmed the overrepresentation of inflammation-related genes among upregulated genes with one third of all genes present in the gene set showing increased expression in the nerves of *Ep400ΔPNS* mice (Supplementary Fig. 6c). This is likely due to increased infiltration, proliferation, and activation of immune cells. Another 6% of upregulated genes were associated with the development or function of skeletal muscle for unknown reason. A contamination by muscle cells appears unlikely. A minor group of genes were associated with nervous system development and will be discussed below. Downregulated genes on the other hand were strongly associated with different aspects of lipid metabolism (Fig. 3c). GSEA confirmed the preferential accumulation of lipid metabolic genes among downregulated genes (Supplementary Fig. 6d). Considering the high demand of lipids for myelin synthesis, this was expected for the *Ep400ΔPNS* mutant. GSEA additionally revealed enrichment of myelination genes among downregulated genes (Supplementary Fig. 6e). This had escaped GO analyses because myelin gene expression in sciatic nerves at P9 is still on the rise and not yet fully induced. Further mining of RNA-Seq data revealed largely unaltered expression of other SC-enriched genes in agreement with qRT-PCR results (Fig. 2l).

To further study alterations in gene regulation, we performed chromatin immunoprecipitation–sequencing (ChIP-Seq) on P9 sciatic nerves. H3K27ac-specific antibodies were used to identify transcribed genes as this histone modification is enriched in active promoters. By sorting genes according to H3K27ac presence around the transcriptional start site (TSS), we established a heatmap for gene transcription in control nerves (Fig. 3d, left). Arrangement of genes in the same order after ChIP-Seq of chromatin from *Ep400ΔPNS* nerves revealed a substantial redistribution of promoter-associated H3K27ac modifications (Fig. 3d, right). In agreement with RNA-Seq studies, GO analyses of the genes that gained H3K27ac at their promoter in *Ep400ΔPNS* nerves showed a strong association with immune and inflammatory responses (Supplementary Fig. 7a). Analogous studies on the genes that lost H3K27ac at their promoter were not informative.

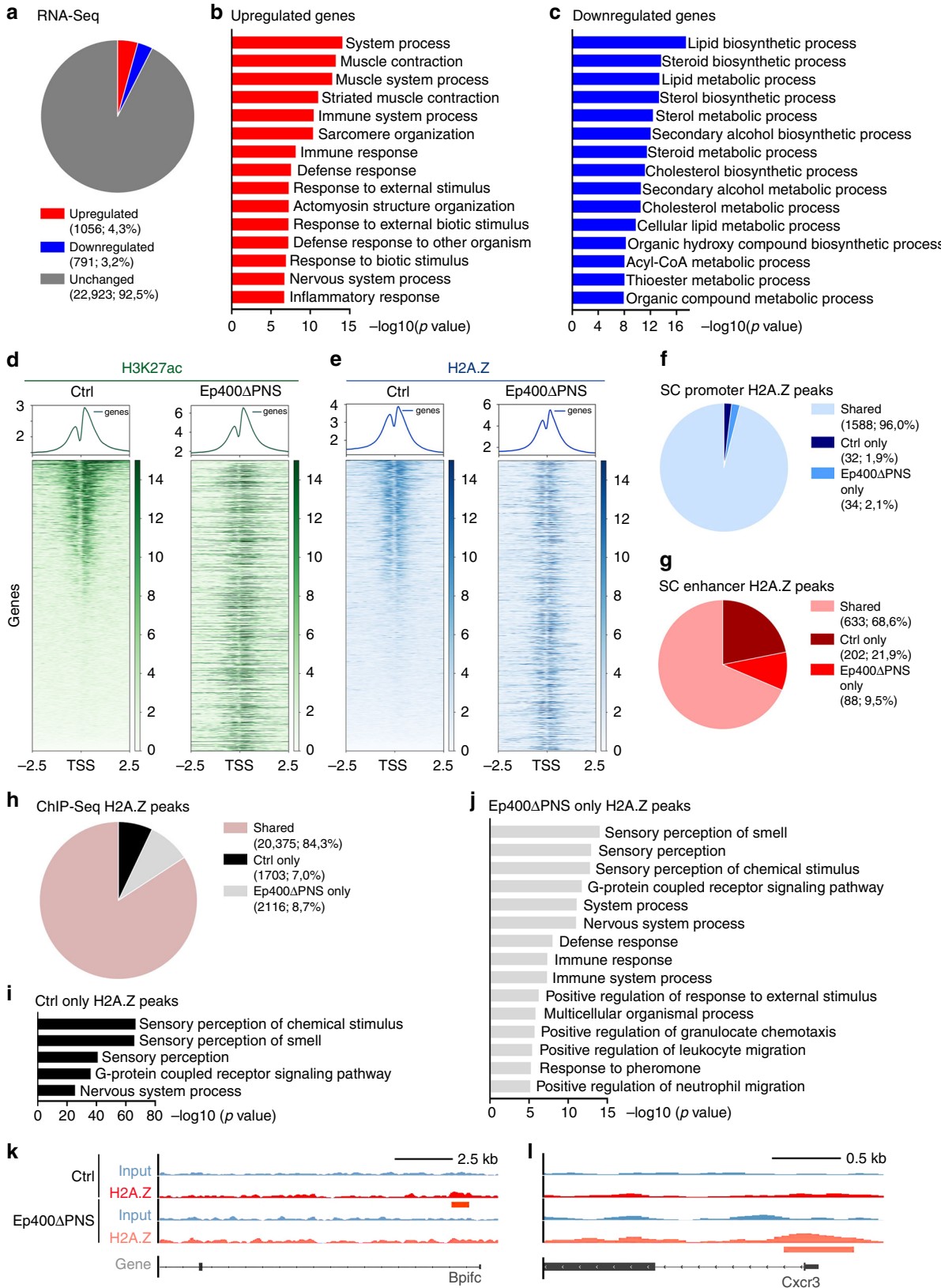

ChIP-Seq was also performed with H2A.Z-specific antibodies (Fig. 3e, for PCA and model-based analysis (MA) blots, see Supplementary Fig. 7b, c). Like the H3K27ac modification, the H2A.Z isoform was enriched around the TSS. As evident from heatmaps of control and *Ep400ΔPNS* mice in which genes were ordered according to H2A.Z enrichment around the TSS in control samples (Fig. 3e), H2A.Z was also substantially redistributed in the mutant.

Among the upregulated genes in *Ep400ΔPNS* nerves that had gained H2A.Z in their promoter, 37% had also gained H3K27ac, whereas 52% were already H3K27ac-positive before. Another 11% had gained H2A.Z in their promoter but remained negative for

**Fig. 3** Expression and H2A.Z-binding profiles in nerves of *Ep400ΔPNS* mice. **a** Pie chart of expression data from RNA-Seq studies on sciatic nerves at P9 depicting genes with ≥2-fold upregulated (red), ≥2-fold downregulated (blue), and unchanged (dark gray) expression in *Ep400ΔPNS* mice relative to controls. **b**, **c** Gene ontology (GO) analysis (GORILLA) for biological processes enriched among genes upregulated (**b**) or downregulated (**c**) in *Ep400ΔPNS* nerves. Processes are sorted by statistical significance. **d**, **e** Heatmaps of H3K27ac (**d**) and H2A.Z (**e**) chromatin immunoprecipitation–sequencing (ChIP-Seq) signals ±2.5 kb around the transcriptional start site (TSS) in control and *Ep400ΔPNS* nerves at P9. The order of genes from top to bottom is according to the strength of the signal for the respective histone modification in the control. **f**, **g** Pie chart showing the fraction of promoters (**f**) and enhancers (**g**) that are active in early postnatal Schwann cells[23] and at the same time occupied by H2A.Z in control nerves (dark blue/red), *Ep400ΔPNS* nerves (medium blue/red), or both (light blue/red). **h** Pie chart depicting H2A.Z peaks selectively detected only in control (black) or *Ep400ΔPNS* (light gray) mice or shared between the samples (rose colored) according to ChIP-Seq studies on sciatic nerves at P9. **i**, **j** GO analysis (GORILLA) for biological processes enriched among genes exhibiting H2A.Z peaks only in control (**i**) or *Ep400ΔPNS* (**j**) nerves. **k**, **l** Selected tracks for genes with H2A.Z peaks only in control (Bpifc, **k**) or *Ep400ΔPNS* (Cxcr3, **l**) nerves. Shown are H2A.Z precipitates and input for both genotypes. Peaks are marked by red (ctrl) and orange (*Ep400ΔPNS*) lines below the respective equally colored tracks

H3K27ac. While this may point to the existence of a small number of H2A.Z-containing, H3K27ac-negative active promoters, more trivial explanations cannot be excluded. Changes in H2A.Z and H3K27ac promoter occupancy were thus partially overlapping in sciatic nerves upon Ep400 deletion but not identical.

To analyze the H2A.Z-specific ChIP-Seq data in more detail, we first focused on changes in H2A.Z occupancy of regulatory regions previously identified as being active in early postnatal SCs[23]. We categorized these regulatory regions as promoters (in TSS vicinity) or enhancers (>1.5 kb away from the TSS). Among the 1654 promoters, 1588 were occupied by H2A.Z in both control and *Ep400ΔPNS* nerves (Fig. 3f). Only 32 were selectively H2A.Z-positive in control nerves and 34 in *Ep400ΔPNS* nerves. GO analysis of the associated genes was not informative because of the low number of genes associated with any particular GO term. A similar picture was obtained for the putative 1693 enhancers where 663 were marked by H2A.Z in both control and *Ep400ΔPNS* nerves, 202 only in control, and 88 selectively in *Ep400ΔPNS* nerves (Fig. 3g). We conclude from these data that the majority of regulatory regions that are active in SCs at early postnatal times retain their H2A.Z status.

To expand our analysis to genes whose regulatory regions are normally not active in early postnatal SCs, we performed a global analysis of H2A.Z peaks. A considerable 84% fraction of the genome-wide H2A.Z peaks were shared (Fig. 3h, Supplementary Fig. 7c). Approximately 7% of the peaks were selectively found in chromatin from control nerves (for example, in the *Bpifc* genomic region coding for BPI fold containing protein C, see Fig. 3k) and 8.7% in chromatin from *Ep400ΔPNS* nerves (for example, in the *Cxcr3* genomic region coding for the C-X-C motif chemokine receptor 3, see Fig. 3l). GO analysis revealed only few biological processes with significant enrichment among the genes with selective peaks in control nerves and was not informative (Fig. 3i). Genes with selective peaks in the nerves of *Ep400ΔPNS* mice on the other hand were frequently associated with immune cell migration, chemotaxis, immune response, and related terms (Fig. 3j) and were also found upregulated in RNA-Seq studies. These peaks are likely contributed by infiltrating and activated immune cells in the mutant nerve.

As the analysis of lost and acquired H2A.Z peaks did not yield insights into the underlying mechanism, we chose to redirect our analysis to alterations in the 84% of H2A.Z peaks that were shared between samples from *Ep400ΔPNS* mice and controls. In samples from *Ep400ΔPNS* mice, 2.1% were reproducibly broader and higher, whereas 3% were narrower and lower (Fig. 4a). The broader and higher peaks were enriched in genes associated by GO studies with developmental processes, including cell fate commitment, specification, differentiation, and organ development as well as with transcription (Fig. 4b), whereas the narrower and lower peaks showed enrichment for genes associated with specific aspects of lipid metabolism (Fig. 4c).

Comparison of the 59 transcription-related genes with broader and higher H2A.Z peaks in samples from *Ep400ΔPNS* mice with those upregulated in RNA-Seq revealed a substantial overlap of 12 transcription factor genes (Fig. 4d). Many of these factors are also associated with developmental processes including differentiation, morphogenesis, and organ development by GO analysis (Fig. 4e) and function in early phases of neural development. Pax3, Pou3f3, Sox1, Sox3, and Tfap2a in particular caught our attention, as these transcription factors are expressed during early phases of SC development (i.e., Pax3 and Tfap2a)[24,25] or are highly homologous to and functionally redundant with transcription factors relevant for SC lineage progression (i.e., Sox1 and Sox3 with Sox2 and Pou3f3 with Oct6). Closer ChIP-Seq data inspection revealed that H2A.Z peak changes in the vicinity of these genes were dramatic, correlated for all transcription factor genes with increased H3K27ac presence, and mostly affected the promoter regions as exemplarily shown for *Tfap2a* and *Pax3* (Fig. 4f, g). At least for *Tfap2a* and *Pax3*, the H2A.Z peaks remained broader and higher in sciatic nerve chromatin of *Ep400ΔPNS* mice at P21 and at 2 months according to ChIP-qPCR studies and appeared permanent (Supplementary Fig. 7d–q and Supplementary Table 13). Therefore, we postulated that failed downregulation and misexpression of these transcription factors impeded SCs in their attempt to myelinate and maintain the differentiated state.

In contrast, H2A.Z and H3K27ac distribution near the TSS of *Krox20*, *Oct6*, and myelin genes such as *Mbp* and *Mpz* exhibited no statistically significant alterations in chromatin from *Ep400ΔPNS* nerves as compared to controls (Supplementary Fig. 8a–d). This suggests that genes coding for regulators of SC differentiation and myelin genes were not primarily affected by Ep400 loss.

**Persistent early transcription factors in Ep400-negative SCs.** To confirm that transcription factors commonly associated with an immature state in neural cells are indeed inappropriately expressed in myelinating SCs of *Ep400ΔPNS* mice, we first analyzed Sox2 occurrence. Sox2 is normally restricted to immature SCs and not found in myelinating SCs as evident from its absence in Krox20-positive cells of control nerves at P21 (Fig. 4h). In contrast, a substantial number of Krox20-positive SCs in age-matched nerves of *Ep400ΔPNS* mice stained positive for Sox2 confirming that expression was not properly shut off in Ep400-deficient SCs. Immunohistochemical findings were corroborated by qRT-PCR (Fig. 4i). In addition to Sox2, transcription factors Tfap2a, Pou3f3, Pax3, Sox1, and Sox3 were dramatically upregulated in postnatal sciatic nerves of *Ep400ΔPNS* mice according to qRT-PCR and in situ hybridization (Fig. 4i, j).

To study the potential effect of their presence on *Krox20* expression and Krox20-dependent myelin gene expression, we co-transfected Neuro2a cells with transcription factor expression

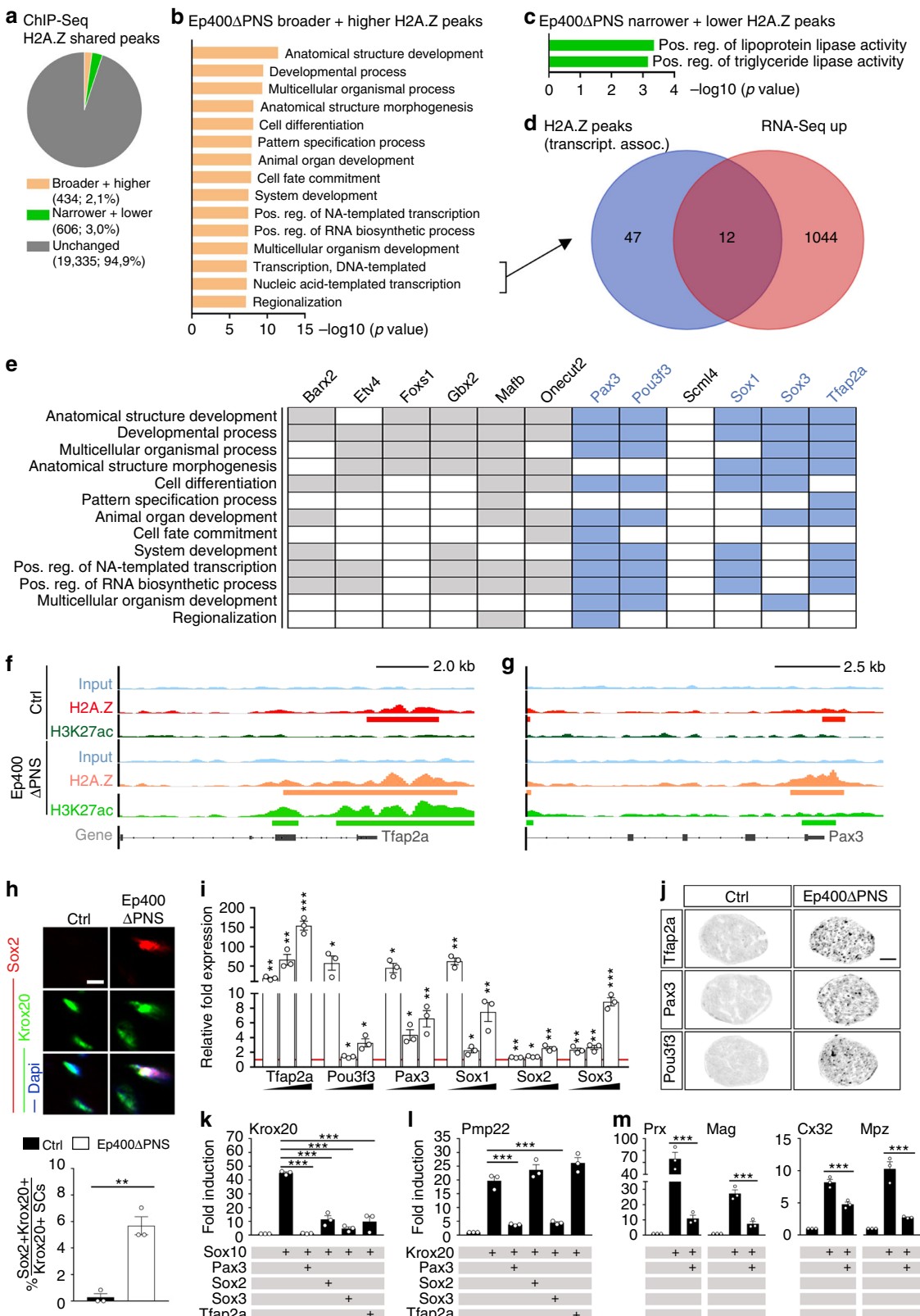

plasmids and luciferase reporters under control of regulatory regions with activity in myelinating SCs. In accord with previous reports[26], the myelinating SC enhancer (MSE) of the *Krox20* gene was robustly activated by Sox10 (Fig. 4k). However, Sox10-dependent activation was severely reduced in the presence of

Pax3, Sox2, Sox3, or Tfap2a arguing that their presence in myelinating SCs may reduce Krox20 levels.

Krox20 itself activates many myelin genes by binding to their regulatory region[22,27]. This includes the *Pmp22*, *Periaxin*, *Mag*, *Connexin32*, and *Mpz* genes[28–32]. Krox20-dependent activation

**Fig. 4** Aberrant H2A.Z occupancy and expression of developmental transcription factor genes in *Ep400ΔPNS* nerves. **a** Pie chart of shared H2A.Z peaks grouped as unchanged (dark gray), broader and higher (ochre) or narrower and lower (green) in *Ep400ΔPNS* nerves at P9. **b, c** Gene ontology terms (GORILLA) enriched among genes with broader and higher (**b**) or narrower and lower (**c**) H2A.Z peaks in *Ep400ΔPNS* nerves, sorted by statistical significance. **d** Venn diagram showing overlap between upregulated genes and transcription-related genes with broader and higher H2A.Z peaks in *Ep400ΔPNS* nerves. **e** Biological processes linked to upregulated transcription factors (blue, further analyzed) with broader and higher H2A.Z peaks in *Ep400ΔPNS* nerves. **f, g** H2A.Z (red), H3K27ac (green), and input (blue) tracks for *Tfap2a* (**f**) and *Pax3* (**g**) genes in control (dark colors) and *Ep400ΔPNS* nerves (light colors) with peaks marked below the tracks. **h** Immunofluorescent visualization and quantification of Sox2 expressing (red) Krox20-positive (green) Schwann cells in control and *Ep400ΔPNS* sciatic nerve sections at P21 (*n* = 3; mean values ± SEM). **i** Reverse transcriptase PCR quantification of *Tfap2a, Pou3f3, Pax3, Sox1, Sox2,* and *Sox3* transcripts in *Ep400ΔPNS* sciatic nerves at increasing age (P9, P21, and 2 months from left to right, indicated by triangle) as compared to age-matched control (arbitrarily set to 1 and marked by red line; *n* = 3; mean values ± SEM; statistics performed between control and *Ep400ΔPNS* mice for each gene and age). **j** In situ hybridization for *Tfap2a, Pax3,* and *Pou3f3* on control and *Ep400ΔPNS* sciatic nerve sections at P21. **k–m** Activation of luciferase reporters under control of *Krox20* (**k**), *Pmp22* (**l**), *Periaxin* (*Prx*), *Mag, Connexin32* (*Cx32*), and *Mpz* (**m**) regulatory regions in transiently transfected Neuro2a cells by Sox10, Krox20, Pax3, Sox2, Sox3, Tfap2a, and combinations (*n* = 3; presented as fold inductions ± SEM, transfections without added transcription factors set to 1 for each regulatory region). Scale bars: 5 μm (**h**), 50 μm (**j**). Statistical significance was determined by unpaired two-tailed Student's *t* test (**h, i**) or analysis of variance (**k–m**) (*$P \leq 0.05$; **$P \leq 0.01$; ***$P \leq 0.001$). Exact values are listed in Supplementary Tables 6–8 and source data are provided as a Source Data file

of these regulatory regions was also sensitive to some of these transcription factors such as Pax3 and refractory to others (Fig. 4l, m). We conclude that misexpression of early transcription factors may affect SC differentiation by interference with expression and/or activity of Krox20.

If the observed defect in myelinating SCs was caused by misexpression of early developmental regulators in *Ep400ΔPNS* mice, phenotypic abnormalities should be rescued by their deletion. To test this assumption, we focused on Tfap2a because this factor exhibited the most dramatic upregulation in *Ep400ΔPNS* mice and broadly influences early neural crest development[33]. Further, because Tfap2a is not expressed beyond the precursor stage in SCs[24] we reasoned that a *Dhh::Cre*-dependent deletion of Tfap2a in *Tfap2aΔPNS* mice should remain without phenotypical consequences on SC differentiation and myelination. In line with our assumption, we did not detect significant changes in SC numbers, marker expression, or myelin parameters in *Tfap2aΔPNS* mice (Fig. 5e–k, Supplementary Fig. 8e).

Mice with a *Dhh::Cre*-dependent deletion of both Tfap2a and Ep400, in contrast, exhibited an unsteady gait and impaired motor coordination as symptoms of a peripheral neuropathy. However, these double knockout (dko) mice were less affected than *Ep400ΔPNS* mice and survived longer (Supplementary Fig. 8f). By immunohistochemical analysis, SCs were present in normal numbers in the sciatic nerve at P0, P9, P21, and at 2 months (Fig. 5a). Compared to control and similar to *Ep400ΔPNS* mice, dko mice still exhibited increased numbers of Sox2-positive cells from P9 onwards and of Oct6-positive SCs at P21, but the increases were less pronounced than in *Ep400ΔPNS* mice (Fig. 5b, c). Strikingly, the number of Krox20-expressing SCs in dko mice was more similar to controls than to *Ep400ΔPNS* mice at P0, P9, and P21 (Fig. 5d) and there was a concomitant increase in Mbp- and *Mpz*-expressing cells (Fig. 5e–m) indicating that the number of myelinating Schwann cells had significantly recovered in dko as compared to *Ep400ΔPNS* mice upon additional Tfap2a deletion. Those myelin sheaths that had formed in dko mice at P21, were thicker, and exhibited a significantly lower mean *g* ratio than the ones in *Ep400ΔPNS* mice, although the value was still higher than in controls (Fig. 5n, Supplementary Fig. 8h, i). The amount of degenerated myelin and myelin outfoldings in sciatic nerve sections of dko mice at P21 were intermediate between control and *Ep400ΔPNS* mice confirming the partial rescue (Fig. 5o, p). In contrast, there was no significant improvement in the average length of node and paranode or regarding the number of Remak bundles in dko mice at P21 (Fig. 5q, r, Supplementary Fig. 8g), although Remak

bundle morphology appeared more normal than in *Ep400ΔPNS* mice (Supplementary Fig. 8h). Numbers of Ki67-positive proliferating and cleaved Caspase 3-positive apoptotic SCs in dko mice were closer to those in controls than in *Ep400ΔPNS* mice (Fig. 5s, t). The number of Iba1-positive macrophages was also reduced in sciatic nerves of dko mice as compared to *Ep400ΔPNS* mice at P9 and P21, although their number remained elevated (Fig. 5u). We conclude that Tfap2a deletion in SCs led to a substantial, though incomplete rescue of most defects caused by the absence of Ep400.

**Discussion**

We show that Ep400 and, by inference, the chromatin remodeling activity of the Tip60/Ep400 complex have a strong influence on terminal differentiation of SCs and myelination. Considering that myelin sheaths and Remak bundles exhibited structural abnormalities after SC-specific Ep400 deletion, both myelinating and non-myelinating SCs appear to rely on Ep400 for differentiation. Ep400 is involved in H2A.Z distribution throughout the genome and simultaneously affects H3.3 positioning[12]. We have confirmed in our study that genomic distribution of H2A.Z is altered after *Ep400* deletion. Concomitant changes in H3.3 distribution are expected but could not be experimentally studied because commercially available H3.3 antibodies did not work in our hands. We conclude from our studies that proper deposition of H2A.Z and likely of H3.3 are a requirement for SC differentiation.

In the early postnatal period of active myelination, *Ep400ΔPNS* mice exhibited strongly reduced numbers of myelin-forming Krox20-expressing SCs and abnormally thin and shortened myelin sheaths as well as abnormalities at the node of Ranvier. Expression of genes coding for myelin proteins or proteins with functions in lipid metabolism was also reduced. After 2 months, myelin thickness as well as the number of myelin sheaths and myelinating SCs had improved. However, the expression levels of genes associated with the myelination process remained reduced at the cellular level. This argues that SCs were still able to myelinate in the absence of Ep400, but did so delayed and did not reach the normal steady state myelin levels. Importantly, H2A.Z (and H3K27ac) distribution in the vicinity of *Krox20*, the major myelin genes, and many other genes with expression in myelinating SCs exhibited little alteration in chromatin from *Ep400ΔPNS* nerves as compared to control arguing that they are not primarily affected by Ep400 deficiency. In contrast, H2A.Z distribution was dramatically and permanently changed in promoter regions of several transcriptional regulators that are

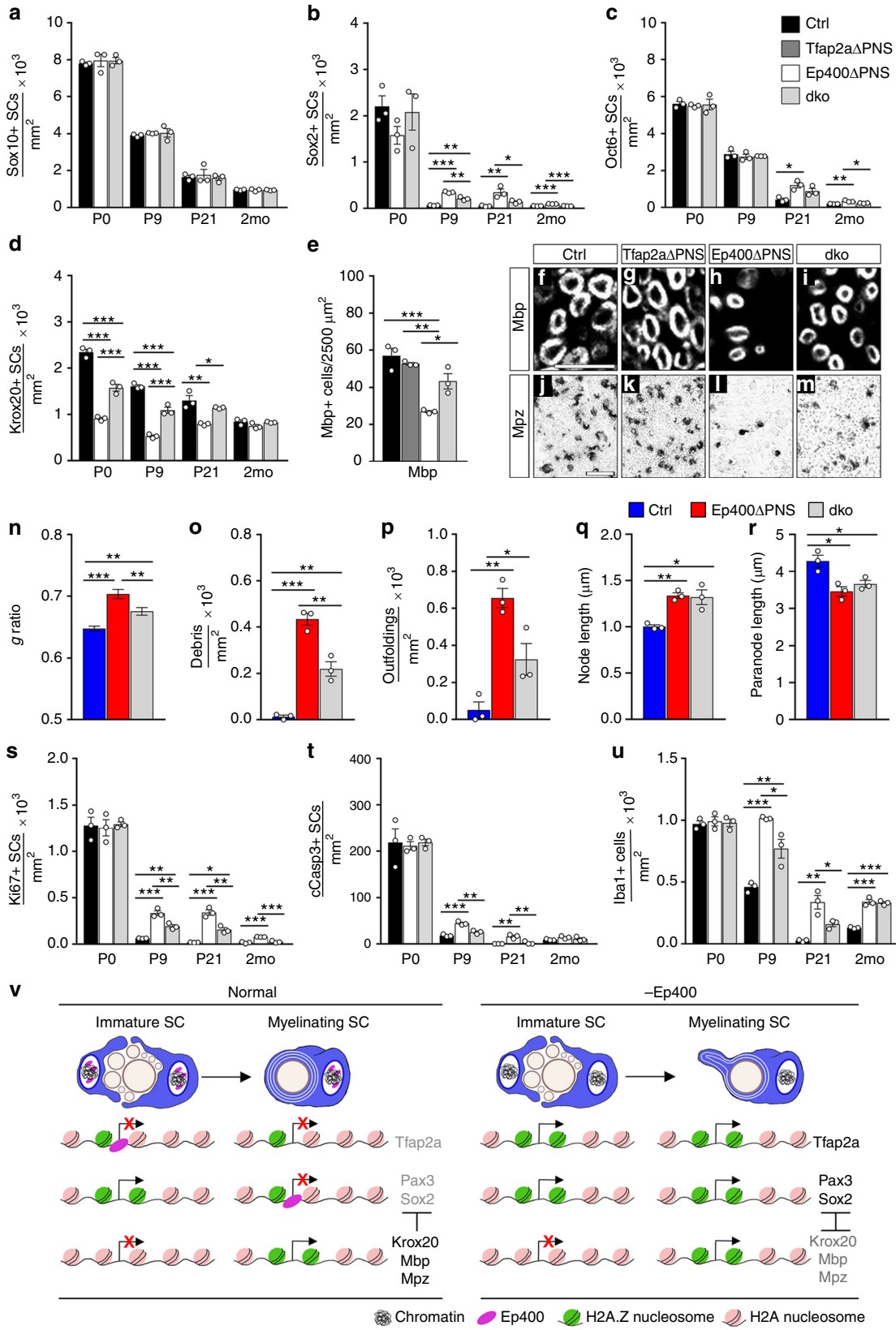

normally active in neural precursors or SCs at early developmental stages (Fig. 5v).

As a consequence, postnatal SCs in *Ep400ΔPNS* mice continued to express these transcriptional regulators beyond their normal window into the differentiating stage (Fig. 5v). Among others, these include Pax3, Sox2, its relatives Sox1 and Sox3, the Oct6-related Pou3f3, and Tfap2a. Tfap2a expression, in particular, is noteworthy as it is normally already turned off during

**Fig. 5** Partial rescue of Schwann cell (SC) defects in *Ep400ΔPNS* mice by Tfap2a deletion. **a–e** Determination of Sox10-positive SCs (**a**) and the number of Sox2- (**b**), Oct6- (**c**), Krox20- (**d**), and Mbp-positive (**e**) subpopulations in sciatic nerve sections of control (black bars), *Tfap2aΔPNS* (dark gray bars), *Ep400ΔPNS* (white bars), and double knockout (dko, light gray bars) mice at P0, P9, P21, and 2 months ($n = 3$; mean values ± SEM). **f–m** Immunohistochemistry with antibodies against Mbp (**f–i**) and in situ hybridization with a *Mpz*-specific probe (**j–m**) on sciatic nerve sections from control, *Tfap2aΔPNS*, *Ep400ΔPNS*, and dko mice at P21. **n–r** Determination of mean g ratio (**n**), myelin debris (**o**), myelin outfoldings (**p**), nodal (**q**), and paranodal (**r**) length from ultrathin sciatic nerve sections of control (blue bars), *Ep400ΔPNS* (red bars), and dko (light gray bars) mice at P21 (mean values ± SEM, $n = 3$ in **o–r** and 100–200 axons in **n**). Values for control and *Ep400ΔPNS* mice are those shown in Fig. 1q and Supplementary Fig. 3s–v. **s–u** Comparison of Ki67-positive proliferating SCs (**s**), cleaved Caspase 3-positive apoptotic SCs (**t**), and Iba1-positive macrophages (**u**) in sciatic nerve sections of control (black bars), *Ep400ΔPNS* (white bars), and dko (light gray bars) mice at P0, P9, P21, and 2 months ($n = 3$; mean values ± SEM). Statistical significance was determined by analysis of variance (*$P ≤ 0.05$; **$P ≤ 0.01$; ***$P ≤ 0.001$). Exact values are listed in Supplementary Tables 1, 4, 5, 9, and 11 and source data are provided as a Source Data file. **v** Model summarizing the proposed action of Ep400. In normal SCs, Ep400 helps to turn off expression of transcriptional regulatory genes active in the precursor (Tfap2a) and immature (Pax3, Sox2) SC stage by replacing H2A.Z-containing by H2A-containing nucleosomes at their transcriptional start site. In Ep400-deficient SCs, genes keep their H2A.Z-containing nucleosomes, remain aberrantly expressed, and interfere with the expression of Krox20 and the myelin genes

embryonic development around the SC precursor stage[24] but was still expressed in the nerves of *Ep400ΔPNS* mice 2 months after birth.

It is easy to envision that prolonged expression of these transcription factors and a failure to shut them off may hinder SCs to initiate myelination, reach normal levels of myelin gene expression, and maintain the differentiated state, making it responsible both for delayed myelination and persistent hypomyelination in the mutant (Fig. 5v).

In fact, both Sox2 and Pax3 have been shown to interfere with myelin gene expression and repress *Krox20* expression[25,34,35]. The current study confirmed that Sox2 and Pax3 directly inhibited Sox10-dependent activation of *Krox20* expression and that Tfap2a acted comparably. Co-expression of Sox2 and Krox20, as observed in this study, is normally not seen in SCs and likely contributes to lower Krox20 amounts and reduced levels of myelin gene expression (Fig. 5v). It may also explain that Ep400-deficient SCs stay longer in a proliferative state. Prolonged co-expression of regulators of the immature state with regulators of differentiation may also trigger an apoptotic response. Thus it is possible that increased rates of proliferation and apoptosis in *Ep400ΔPNS* mice are a direct consequence of the continued expression of transcription factors that are normally only observed in immature cells. However, we cannot rule out that the relationship is more complicated. Apoptosis could, for instance, cause a compensatory upregulation of proliferation rates in mutant SCs.

Increased apoptosis may also be responsible for the higher number of activated macrophages and the immune cell response signature in sciatic nerves of *Ep400ΔPNS* mice. More macrophages have also been observed in a mouse model with continuous Sox2 overexpression in SCs[35]. In comparison to this mouse model, Sox2 and other transcriptional regulators are probably expressed at lower levels in *Ep400ΔPNS* mice, which may explain the milder phenotypic defects.

To test our hypothesis that persistent expression of regulators of the immature state contributes causally to the observed SC phenotype, we deleted Tfap2a as one of the deregulated transcription factors in SCs of *Ep400ΔPNS* mice and achieved a remarkable rescue in the resulting dko mice.

While this is the first study that addresses the role of histone-exchanging chromatin remodelers and variant histones in SCs, other chromatin remodelers with nucleosome removal, addition, and repositioning activity have previously been studied. By deleting the central ATP-hydrolyzing Brg1 and Chd4 subunits, both the SWI/SNF-type BAF complex and the CHD-type NuRD complex have been demonstrated to be important for terminal differentiation of SCs[36–39]. Whereas defects were severe and permanent in case of the Brg1-containing BAF complex, they were milder and transient in case of the Chd4-containing NuRD complex. Effects of the Brg1-containing BAF complex were ascribed to its ability to activate Sox10 expression in a neuregulin- and nuclear factor-κB-dependent manner, to interact with Sox10 in the induction of Oct6 and Krox20 as central regulators of late SC development, and to directly stimulate the expression of myelin genes[37–39]. In case of the Chd4-containing NuRD complex, most effects appeared downstream of Krox20 and its Nab cofactors with which it interacts to activate myelin gene expression and repress inhibitors of the myelination process[36]. Comparison of the phenotypic defects and underlying mechanisms in the respective mutants with the ones in *Ep400ΔPNS* mice shows that different chromatin remodeling activities are required for SC differentiation, but each influence different aspects during peripheral myelination.

In summary, our study shows for the first time that histone-exchanging chromatin remodeling activity is important for terminal differentiation of SCs because it helps to shut off transcriptional regulators of the immature state and thereby promotes differentiation and stabilizes the differentiated state. Interestingly, transcriptional regulators of the immature state were found increased in patients suffering from Charcot–Marie–Tooth disease and mouse models of this peripheral neuropathy[40–42]. While the consequences of this increase are complex, they point to relevance of our findings for human disease. They also argue that modulation of the remodeling activity of the Ep400/Tip60 complex may be useful in disease treatment.

## Methods

**Transgenic mice**. Mouse housing and experiments were in accordance with animal welfare laws and all relevant ethical regulations and were approved by the responsible local committees and government bodies (Veterinäramt Stadt Erlangen, Regierung von Unterfranken). The *Ep400^{fl/fl}* mouse strain (RBRC03987) was provided via RIKEN BRC through the National Bio-Resource Project of MEXT, Japan. To delete Ep400 in developing neural crest or SCs, floxed alleles for *Ep400*[14] were combined in mice with a *Sox10::Cre* Bac transgene[16,43] or a *Dhh::Cre* transgene[17] to yield *Ep400ΔNC* and *Ep400ΔPNS* mice. In some *Ep400ΔPNS* mice, a floxed *Tfap2a* allele[44] was additionally crossed in. Genotyping was as described[14,16,17,44]. Transgenic and control mice were on a mixed C3H x C57Bl/6J background and kept with continuous access to food and water under standard housing conditions in 12:12 h light–dark cycles. For bromodeoxyuridine (BrdU) or 5-ethynyl-2′-deoxyuridine (EdU) labeling, 100 μg BrdU (Sigma, #B5002) or EdU (Invitrogen, #A10044) per gram body weight were used. Mice were injected intraperitoneally with EdU 24 h and with BrdU 1 h before tissue preparation. Embryos were obtained at E12.5, E14.5, E16.5, and E18.5 and sciatic nerves at P0, P9, P15, P21, and at 2 months of age from male and female mice with relevant genotypes.

**Tissue stainings**. Immunohistochemistry and in situ hybridization were performed on sections of the trunk region of mouse embryos and the sciatic nerve or

on teased fibers from sciatic nerves[45]. Tissue fixation in 4% paraformaldehyde, transfer to 30% sucrose, and freezing in Tissue Freezing Medium (Leica)[32] was followed by preparation of 10-μm transverse cryotome sections. In situ hybridization was performed with DIG-labeled antisense riboprobes specific for *Tfap2a* (positions 353–2056 acc. to accession number BC018226), *Pou3f3* (positions 69–1582 acc. to accession number NM_138837), *Pax3* (positions 396–1839 acc. to accession number NM_053710), and *Mpz* (positions 1–1864 acc. to accession number NM_001314068). The following primary antibodies were used in immunohistochemical stainings: rat anti-MBP monoclonal (Bio-Rad, #MCA409S, Lot #210610, 1:500 dilution), rat anti-Pecam antiserum (Pharmingen, #550274, Clone #MEC13.3, 1:50 dilution), rabbit anti-Caspr antiserum (Abcam, #AB34151, Lot#GR86230, 1:1000 dilution), guinea pig anti-Sox10 antiserum (homemade, validated on control and knockout mouse tissue, 1:1000 dilution)[46], rabbit anti-Ep400 antiserum (homemade, generated against a bacterially expressed and purified peptide corresponding to amino acids 507–661 of mouse Ep400 according to accession number NP_083613, validated on control and knockout mouse tissue, 1:1000 dilution), guinea pig anti-Krox20 antiserum (homemade, generated against a bacterially expressed and purified peptide corresponding to amino acids 28–166 of mouse Krox20 according to accession number NM_010118.3, validated on mouse sciatic nerve with and without Krox20 expression, 1:1000 dilution), rabbit anti-Sox2 antiserum (homemade, generated against a bacterially expressed and purified peptide corresponding to amino acids 10–38 of mouse Sox2 according to accession number NM_011443, validated on control and knockout mouse tissue, 1:500 dilution), rabbit anti-Oct6 antiserum (homemade, generated against baculovirus-expressed and purified full-length mouse Oct6 according to accession number NM_011141.2, validated on control and knockout mouse tissue, 1:2000 dilution), rabbit anti-Nav1.6 antiserum (Alomone Labs, #ASC-009, Lot#AS-C009AN2425, 1:50 dilution), rabbit anti-Iba1 antiserum (Wako, #019–19741, Lot#SAE6921, 1:250 dilution), rabbit anti-cleaved caspase 3 antiserum (Cell Signaling Technology, #9661, Lot#0043, 1:200 dilution), rabbit anti-Ki67 antiserum (Thermo Fisher Scientific, #RM-9106, Lot#9106S906D, 1:500 dilution), rabbit anti-CD3 antiserum (Abcam, #ab5690, Lot#665620, 1:500 dilution), rabbit anti-Desmin antiserum (Abcam, #ab15200, 1:1000 dilution), rabbit anti-fibronectin antiserum (Abcam, #ab2413, Lot#GR3235936–2, 1:100 dilution), and mouse anti-α-smooth muscle actin antiserum (Sigma, #A5228, 1:200 dilution). Secondary antibodies were coupled to Cy3 (Dianova, 1:200 dilution), Cy5 (Dianova, 1:200 dilution), or Alexa488 (Molecular Probes, 1:500 dilution) fluorescent dyes. Nuclei were counterstained with DAPI. Incorporated EdU was visualized by the Click-iT EdU Alexa Fluor 488 Imaging Kit (Invitrogen, #C10337) and incorporated BrdU by rat anti-BrdU monoclonal (Abcam, #ab6326, Lot#GR3173637–7, 1:100 dilution). TUNEL was performed according to the manufacturer's protocol (Chemicon). Paraphenylene-2,6-diamine staining of sciatic nerve sections was as described[47]. Stainings were documented with a Leica DMI6000 B inverted microscope equipped with a DFC 360FX camera or a Leica DMIRE2 confocal microscope (all Leica).

**Electron microscopy.** Sciatic nerves of control and genetically altered mice were dissected at P21 and at 2 months of age, followed by fixation in cacodylate-buffered fixative containing 2.5% paraformaldehyde and 2.5% glutaraldehyde, incubation in cacodylate-buffered 1% osmium ferrocyanide, dehydration, embedding in Epon resin, transverse or longitudinal sectioning, and staining with uranyl acetate and lead citrate. Fifty-nm sections were examined with a Zeiss Libra electron microscope (Carl Zeiss, Inc.). From electron microscopic pictures, the number of myelinated axons >1 μm and the *g* ratio of myelinated axons were determined.

**RNA-Seq, ChIP-Seq, ChIP-qPCR, and bioinformatic analysis.** Total RNA was prepared from sciatic nerves obtained at P9 from control and *Ep400ΔPNS* mice. RNA samples were treated with DNase I to remove contaminating DNA. Quality and purity of samples were evaluated using an Agilent 2100 Bioanalyzer (Agilent Technologies UK). Four hundred nanograms were used for library preparation (Ilumina Stranded mRNA Kit). Approximately 100 million reads per library were sequenced on an Illumina Hiseq 2500 platform (Next Generation Sequencing Core Facility, FAU Erlangen-Nürnberg) and aligned to mouse genome mm10 using STAR (version 2.5.1. b). Unique mappings to Ensembl Genes (Version 85) were counted using HTSeq. Differential expression analysis was carried out using the DESeq2 package (version 1.10.1) with default parameters in the R statistical environment. Gene expression values are deposited in GEO under accession number GSE119132.

GO analysis of genes upregulated and downregulated in SCs of the *Ep400ΔPNS* sciatic nerve at P9 was performed using the GO enrichment, analysis, and visualization tool (GORILLA, http://cbl-gorilla.cs.technion.ac.il/) or the Database for Annotation, Visualization and Integrated Discovery (DAVID, https://david.ncifcrf.gov/) with comparable results. In most cases, results from GORILLA are shown. GSEA from the Broad Institute was further used to determine whether the defined set of genes shows statistically significant, concordant differences between RNA samples from control and *Ep400ΔPNS* sciatic nerves (http://software.broadinstitute.org/gsea/index.jsp).

In addition, sciatic nerve chromatin underwent crosslinking in 1% formaldehyde and shearing to fragments of approximately 100–250 bp in a Bioruptor (Diagenode). Sheared, quantified, and pre-cleared chromatin was incubated with rabbit antiserum against H3K27ac (Diagenode, #C15410196, Lot#A1723–0041D, 1 μg/IP), rabbit antiserum against H2A.Z (Diagenode,

#C15410201, Lot#A2039P, 1 μg/IP), or control rabbit IgG (Sigma-Aldrich, #PP64, Lot#070M768V, 1 μg/IP) and precipitated using bovine serum albumin-pretreated protein A sepharose beads. Crosslinks were reversed in an aliquot of input and the precipitated chromatin. DNA was purified by proteinase K treatment, phenol/chloroform extraction, and ethanol precipitation, and then used for standard qPCR[48] with specific primers from the *Tfap2a* and *Pax3* genomic regions (see Supplementary Tables 12 and 13) or for library generation with the ChIP-Seq Sample Prep Kit (Illumina). From these libraries, approximately 24 million reads were generated per sample in ChIP-Seq experiments. Data are deposited in GEO under accession number GSE119194.

Reads were mapped to the mouse reference genome (GRCm38/mm10) using Bowtie (version 2.3.4.2). After removing PCR duplicates with RmDup (version 2.0.1), mapped reads were converted to BAM format via SAMtools (version 1.1.2). BAM files were further processed using the BamCoverage tool within the deeptools software (version 3.0.2.0) and normalized to 1×. Experiments with anti-H2A.Z antibodies were performed in biological quadruplicates. ChIP-Seq with anti-H3K27ac antibodies were performed in duplicates. For peak calling, MA of ChIP-Seq (MACS2) (version 2.1.1.20160309.0) was used with input samples set as background and default parameters (*q* value <0.05). Single gene profiles were visualized in the Integrated Genome Viewer (IGV; version 2.3.98). Heatmaps and plotprofiles were created with the deeptools software. For illustrating average profiles over all genes, the computeMatrix scale region mode (version 2.5.0.0) was used by averaging genes with 2.5 kb upstream and downstream of the TSS. To detect ChIP-Seq peaks with differences in signal intensity in the control and *Ep400ΔPNS* condition, differential binding analysis of ChIP-Seq peak data was performed for all called peak regions identified with MACS2 in control and *Ep400ΔPNS* conditions using DiffBind (version 2.6.6). Gene coordinates were downloaded from the UCSC Table Browser (UCSC genes GRCm38/mm10) and used to create a genome-wide promoter list (GWPL, defined as all regions 1.5 kb around the TSS).Differential H2A.Z enrichment at gene promoters was determined by comparing the enriched or depleted peak regions in the *Ep400ΔPNS* condition (*p*-value < 0.05) with GWPL using intersect (version 1.0.0). To detect ChIP-Seq peaks with differences in peak width in the control and *Ep400ΔPNS* condition (broader and higher peaks were defined as peaks at least 1.5 kb broader and/or with at least 1-fold higher signal intensities, narrower and lower peaks as peaks at least 1.5 kb narrower and/or with at least 1-fold lower signal intensities); all called peak regions identified by MACS2 in the control and *Ep400ΔPNS* conditions were separately combined with GWPL using join (version 1.0.0). Comparison of peak regions in control and *Ep400ΔPNS* conditions using the Venn diagram tool of the PBE homepage (http://bioinformatics.psb.ugent.be/webtools/Venn/) generated 3 different gene lists: genes shared in control and *Ep400ΔPNS* conditions, genes with peaks only in the *Ep400ΔPNS* condition, and genes with peaks only in the control condition. The list of genes shared in control and *Ep400ΔPNS* conditions was subsequently divided in genes with broader and higher peaks in the *Ep400ΔPNS* condition, genes with narrower and lower peaks in the *Ep400ΔPNS* condition, and genes with unaltered peaks between control and the *Ep400ΔPNS* condition. For further analysis, the lists of genes enriched in the *Ep400ΔPNS* condition and containing broader and higher peaks were merged, as were the list of genes depleted in the *Ep400ΔPNS* condition and containing narrower and lower peaks. GO analysis of genes differently occupied by H2A.Z in the control and *Ep400ΔPNS* condition was performed using GORILLA or DAVID with comparable results.

**qRT-PCR analysis.** For qRT-PCR, total RNA was reverse transcribed and subjected to PCR on a CFX96 Real-Time PCR System (Bio-Rad) using the primers listed in Supplementary Table 6. The melting curve of each sample was measured to ensure specificity of the amplified products. All samples were processed as technical triplicates. Data were analyzed by the ΔΔCt method.

**Luciferase assays.** Mouse Neuro2a neuroblastoma (obtained from ATCC, authenticated by PCR) were grown in Dulbecco's modified Eagle's medium supplemented with 10% fetal calf serum and transfected with various combinations of luciferase reporter and effector plasmids using Superfect reagent (Qiagen)[48]. In pTATA-luc or pGL2 (Promega) based plasmids, luciferase reporters were under control of SC-specific regulatory regions from the *Krox20* (MSE), *Pmp22*, *Periaxin*, *Mag*, *Connexin32*, and *Mpz* genes[26,28–32]. The pCMV5-based effector plasmids carried open-reading frames for Sox10, Krox20, Tfap2a, Pax3, Sox2, and Sox3[49]. An amount of 0.5 μg was used for each reporter plasmid and 0.1 μg of effector plasmid per well on a 24-well plate. Overall amounts of plasmid were kept constant by adding empty pCMV5 where necessary. Whole-cell extracts were prepared 48 h posttransfection and luciferase activities were determined by detection of chemiluminescence as described[48].

**Quantifications and statistical analysis.** Results from independent animals, experiments, or separately generated samples were treated as biological replicates. Sample size was $n \geq 3$ for all molecular biology experiments and experiments with mice as common for this kind of study except for ChIP-Seq studies with anti-H3K27ac antibodies ($n = 2$). No data were excluded from the analysis. Randomization was not possible. Investigators were not blinded in animal experiments. GraphPad Prism7 (GraphPad software, La Jolla, CA, USA) was used to determine

whether differences in cell numbers, *g* ratios, transcript levels, or immunoprecipitated DNA were statistically significant by analysis of variance or unpaired, two-tailed Student's *t* tests (*$P \leq 0.05$; **$P \leq 0.01$, ***$P \leq 0.001$). The data met the assumptions of the chosen test. Variance between statistically compared groups was similar.

**Reporting summary**. Further information on research design is available in the Nature Research Reporting Summary linked to this article.

## Data availability
All data generated or analyzed during this study are included in this published article, its supplementary information and Source Data file or were deposited in GEO under accession numbers GSE119132 and GSE119194.

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

## Acknowledgements
We thank Dr. D. Meijer (Edinburgh) and Dr. W.D. Richardson (London) for mutant mice. M. Schimmel, D. Jaegers, and T. Baroti are acknowledged for their technical expertise and Dr. S. Johnsen and Dr. Z. Najafova (both Göttingen) for help with bioinformatics. This work was funded by the DFG (We1326/14, GRK2162 to M.W.) and the NIH (DE12728 to T.W.).

## Author contributions

M.W. conceived and supervised the study. F.F. and M.W. designed the experiments. F.F., E.S., and E.R.T. performed the experiments with the help of A.-L.S. and S.H.; T.J.W., T.F., and R.F. provided important reagents and materials. F.F., E.S., E.R.T., and M.W. analyzed the data. F.F. and M.W. wrote the manuscript.

## Additional information

**Competing interests:** The authors declare no competing interests.

