## [Peer Review File · Nature Communications]

Reviewers' comments:

Reviewer #1 (Remarks to the Author):

In the manuscript "Ep400 deficiency in Schwann cells leads to persistent expression of early developmental regulators and peripheral neuropathy", Fröb et al. investigate the role of the chromatin remodeler Ep400 in Schwann cell development. The authors found that the conditional deletion of Ep400 in this lineage leads to peripheral neuropathy, by preventing the downregulation of transcription factors involved in early Schwann cell development. These findings are very interesting and are in line with recent findings that abnormal expression of early regulators in mature neural cells can have pathological consequences. The authors are very thorough in their analysis and I deem that the paper should be accepted to publication in Nature Communications, pending that the authors address the following points:

- The authors focus on the deposition of H2A.Z by EP400, but this chromatin remodeler has been also to be involved in the deposition of the histone variant H3.3, which is also involved in transcriptional activation (Pradhan et., Mol Cell 2016). Given that the authors also observe effects on H3K27ac, the role on H3.3 might be even more relevant. Additional experiments assessing the exact genomic location of Ep400 (by ChIP-Seq) in the same samples where H2A.Z and H3K27ac was analysed and the effects of Ep400 knockdown on H3.3 deposition would be of interest. Nevertheless, given the dramatic phenotype the authors observe upon cKO of Ep400 and that they already provide mechanistic insights for this phenotype, I consider that these experiments are not absolutely required for the current manuscript to be accepted and can be investigated in follow up studies. They should nevertheless be discussed in the paper.

- The authors observe a redistribution of the H3K27ac peaks (Figure 3d) and also of H2A.Z peak (Figure 3e) in the cKO:

- o The white-to-green scales in the 2 panels of Figure 3d should be the same, so the graphs can be directly comparable; the same applied to the blue scales in Figure 3e

- o There are a substantial number of genes that appear to gain H3K27ac in the cKO in Figure 3d. What are these genes? Have the authors performed gene ontology on these?

- o Also, there are a substantial number of genes that appear to gain H2A.Z, but these appear to maintain their H3K27ac status (Figure 3e, panels to the right), so they are most likely not the same genes observed in Figure 3d to gain H3K27ac. The authors mention that these genes are upregulated by RNA-Seq analysis, but does this happen through a H3K27ac independent mechanism? The authors should elaborate on this point.

- The authors have focused their ChIP-Seq analysis at TSSs, and thus promoters. However, H3K27ac also associates with enhancers, and H2A.Z as also been shown to be depleted at enhancers upon EP400 knockdown. The authors should also perform an analysis at the enhancer level.

- In page 10, the authors mention "Arrangement of genes in the same order after ChIP-Seq of chromatin from Ep400 PNS nerves revealed a substantial redistribution of the promoter-associated H3K27ac modifications (Fig. 3d, right) indicating that RNA-Seq likely underestimates the occurring changes in transcription at the cellular level, because RNA-Seq determines transcript levels as an average over all cell types in the nerve." The ChIP-Seq was performed also at a bulk level, so it would entail similar challenges, so I recommend the authors to reformulate the last part of the sentence.

- Do the 12 transcription factors up-regulated in the RNA-Seq and increased H2A.Z peaks exhibit higher levels of H3K27ac? Since the authors focus on this histone modification early in the manuscript, they should comment on it.

- In figure 1r and v, the authors state that statistical significance was determined by two-tailed Student's t-test, which is not the correct test to be used, since the data appears to be normalized to ctrl, which then does not present any variance. This would explain why the observed statistical significance for a 2.85% difference in figure 1v, which is unlikely to have biological relevance. The same applies to figure 2l, where an ANOVA analysis without normalization would be more advisable. The authors should also state the number of ns in all the figures.

- In Figure 2e-k, the authors observed an increase in Oct6+ promyelinating Schwann cells, but then observe a dramatic decrease in Krox20+ myelinating cells. They also observe increased proliferation and cell death and an increase in microglia. While the conclusions of the authors that "Schwann cell differentiation in Ep400 PNS mice is disturbed early during myelination. This is accompanied with a slightly increased rate of apoptosis that is compensated by a similarly increased rate of proliferation." are plausible, these conclusions cannot be directly taken from the data, since the authors present quantifications for proliferation and apoptosis in general without correlating with the cell type. As such, the authors should show countings of Ki67+/ Sox10+ cells, cCasp3+/Sox10+ and Ki67+/ Iba1+ cells, cCasp3+/Iba+.

- The authors should add error bars in Figure 2b

- The authors perform RNA-Seq in the sciatic nerve as bulk, which lead to a significant immune signature that could have masked the effects on Schwann cells. Ideally the authors should have done instead RNA-Seq on Sox10+ Schwann cells. While this would have significantly improved the paper, the authors did a very thorough and persistent analysis of their data, which allow them to identify the changes in early regulators, which they elegantly show later in the paper that are contributors to the observed phenotype.

- In Figure 4g, the authors show a graph with the quantification of %Sox2/Sox10, but which is very informative, but this is not mentioned in the Figure legend, where only the double staining Sox2 and Krox20 is mentioned.

- The authors mention that "these dko mice (Tfap2a and Ep400) were less affected than Ep400 PNS mice in their ability to move in the cage and the majority was still alive at five months of age." A survival curve as in Supplementary Figure 2 would be advisable.

- What statistical test was done for Figure 5b? It is not possible to have statistical significance in Oct6 between EP400delta PNS and dko, the difference is minimal and the variation between the individual points too large

- In Figure 5l, ANOVA should be used for statistical testing

Reviewer #2 (Remarks to the Author):

In this manuscript, Fröb et al. studied the functions of the chromatin-remodeling complex Tip60/Ep400 in Schwann cell development. This complex allows the exchange of the histone H2A by the histone variant H2A.Z in nucleosomes. Fröb et al. employed mouse genetics to specifically ablate in Schwann cell precursors the ATPase subunit Ep400, without which the remodeling complex is not active. By using this approach, the authors show an important function of this chromatin-remodeling complex in postnatal myelination. In addition, they identify the transcription factor Tfap2a as a major target of the Tip60/Ep400 complex by showing that it is strongly upregulated in Ep400 KO Schwann cells and that the ablation of Tfap2a in Schwann cells in addition to Ep400 rescues partially the myelination defect of Ep400 KO Schwann cells. Overall this study is interesting, showing important functions of the chromatin-remodeling complex Tip60/Ep400 in postnatal myelination by the regulation of a set of specific target genes. However,

I have a few comments below to strengthen the study.

1/ Figure 1d: What is the identity of Sox10-negative cells expressing Ep400 in Ep400 Δ PNS sciatic nerves? These cells seem much more abundant in KO nerves. Are they macrophages? Please, add a double staining of Ep400 with a macrophage marker and DAPI labeling.

2/ Page 5: "Ep400 Δ NC mice exhibit multiple neural crest-related abnormalities..."

Please, either explain what type of neural crest-related abnormalities are observed and document these observations or remove the part on neural crest cells.

3/ Is it really necessary to write in the figure legends all numerical values? It renders legends very difficult to read. Why not instead adding an excel file in supplementary data to show all these values, if necessary?

4/ Fig. 2b: Has the relative distribution of internodal length been calculated only for 1 control and 1 Ep400 Δ PNS? It should be done on sciatic nerves of at least 3 animals for each group. Please, add the required number of animals for this analysis, and add error bars and statistics.

5/ Fig. 2c-d: It would be more accurate to measure nodal and paranodal lengths by electron microscopy rather than by immunofluorescence of nodal and paranodal components. An average of 1 μ m for the nodal length found in KO is a value usually found in control animals. Measuring these parameters on longitudinal electron microscopy images would lead to more accurate values and strengthen this point.

6/ Fig. 2h-k: representative photos for KI67, TUNEL, cCasp3 and Iba1 should be added in supplementary figure, such as it is already done in Suppl. Fig.3 for Sox10, Oct6 and Krox20.

7/ Fig. 4g: I do not understand the bar graph under the co-staining Sox2/Krox20... The labeling seems wrong. In addition, values on the y axis cannot be a ratio and do not seem either to be a percentage. Please, correct.

8/ In dko nerves, myelination is improved as compared to Ep400 Δ PNS. What about nodal and paranodal lengths, proliferation and apoptosis? In addition, the g ratio for Tfap2a Δ PNS does not appear on the figure. Please, add these data.

Reviewer #3 (Remarks to the Author):

This is a potentially interesting manuscript on the role of Tip60/Ep400 epigenetic modulator in Schwann cell development, which includes the phenotypic analysis of mice with genetic ablation using the Dhh Cre line and the molecular analysis of chromatin (isolated from sciatic nerves and immunoprecipitated with antibodies specific for the histone variant H2Az and for H3K27ac histone)

The proposed model is that in the absence of Ep400, H2Az persists on chromatin (as broader and higher peaks), thereby leading to inappropriate repression of transcription factors impact differentiation of Schwann cells and the authors support it by providing evidence of partial rescue by ablating also one of these factors (Tfap2a)

MAJOR POINTS

- while the phenotypic analysis and the molecular characterization individually reveal some interesting findings, it is unclear how the two are related. This is a serious concern, as the authors declare that the rationale for the molecular analysis is the identification of the "cause of the phenotype". Unfortunately, there is little attempt to connect molecular results to phenotype, with the exception of focusing on the 12 transcription factors, identified at the intersection of the "broad Chip peaks in mutants" with the "upregulated transcripts".

1. the phenotype appears to include multiple aspects of SC function and needs to be better characterized. there is not only hypomyelination at day P21 (as shown in Fig.1k and quantified in Fig 1q), but also myelin outfoldings(evident in Fig 1o and in Fig, 1x) and aberrant remak bundles. As presented the data are confusing and qualitatively presented. A thorough quantification of each finding (i.e. percentage of myelinated axons, number of remak bundles, axons with myelin outfoldings, axons with myelin degeneration) should be included for earlier (P7, P14, P21) and later (2mo, 6 mo).

2. the findings on the "strong alterations" of the node of Ranvier should be better presented and explained. The inclusion of sagittal ultrastructural images may provide insights. It is unclear whether the molecular analysis provides candidates that are able to explain these findings.
3. the choice of the time points for RNA Seq and Chip Seq is questionable. The authors describe a reduced number of Krox20 cells already at birth, and yet the analysis is conducted much later, thereby reflecting not only changes in gene expression that may be directly related to gene ablation (and therefore "causal") but also many transcriptional changes that are the result of the cellular changes caused by gene ablation (and therefore "consequential"). Earlier analysis of transcripts may provide a better sense of what is causal. An example is the GO "immune response", this is clearly a reactive response, as pointed out by the authors, and it does not shed light on the causal aspect of the phenotype. It is also unclear whether Ep400 regulates different genes at different times of development and that the phenotypes described at p21 is the results of a cumulative number of defects from distinct stages of SC development. At the very minimum, it would be important to mine the RNA Seq data for stage-specific genes and validate differential transcripts at multiple time points.
4. If there is already a problem at birth, why did the authors omit the P0 time point from panels 2h to 2k? Wouldn't be important to understand what are the first changes detected in these mice?
5. The description of the RNA Seq data on page 9 needs to be substantially improved to include the total number of differential transcripts and a brief discussion of the analysis
6. the selection of the Gorilla software for GO analysis is questionable, especially since the authors continue to reference it throughout the paper. The authors may get better highlights from a GSEA analysis of the RNA Seq and selective relevant datasets. Similarly, for the ChipSeq analysis, the selected categories fail to shed light on the phenotype and I wonder whether a more in depth analysis, comparing multiple programs, including DAVID, may provide more meaningful GO categories.
7. the description of the ChipSeq data on page 11 needs to be reorganized and presented in a clearer manner. IGV curves need to be clearly presented in Fig 4 at multiple time points and, most importantly, to address the issue of causality, the ChIP data need to be validated in chromatin samples isolated from nerves at multiple time points (p9, p21 and 2 mo) and immunoprecipitated with antibodies for H2AZ and H3K27Ac from controls and mutants. Chip PCR and IGV curve at multiple time points are truly critical in Fig. 4 to support the main hypothesis of the paper, which in the absence of ep400, peaks of H2Az, which would be lost in the wild type mice, remain in the mutant.
8. While the data on the "partial rescue" are of potential interest, they are truly incomplete. A time-course of proliferation, apoptosis and ultrastructural characterization should be included. Without these data is unclear what the actual "rescue" is due to. There are clearly multiple aspects of SC biology that are impacted by Ep400 and the authors have the opportunity to shed some light on this important question.
9. There is an important inconsistency between the description of the phenotype of the Ep400 mutants, where "mostly large caliber axons appear to be affected" and the rescue in the triple mutant lacking also Tfap2a. It would be important to clarify.

minor points:

- the authors should explain the rationale for focusing on this molecule. Why was it selected? is there a translational relevance due to mutations of the gene in peripheral neuropathies?
- data obtained using the Sox10 cre line are briefly mentioned and not shown, due to the extensive repercussions on neural crest cell development. They may be included.
- the definition of "bigger" peaks is a bit imprecise and would be best if replaced with the same terms used in the methods "broader and higher". The same applies to "smaller" peaks (they are "narrower")
- the text of the manuscript contains a large amount of grammatical errors with present and past tenses incorrectly used (i.e. results should be all in past tense), colloquialisms ("if anything there was..." "goes along with ..."), unusual choice of terms (i.e. "disturbed" differentiation, "higher" age,

"almost normalized"...), repetitions ("normally"... "normally" within the same sentence) and this is truly distracting for the reader. The text needs to be extensively revised.

Point by point response to the reviewers' comments

on manuscript NCOMMS-18-30213

by Fröb et al.

"Ep400 deficiency in Schwann cells leads to persistent expression of early developmental regulators and peripheral neuropathy"

We were pleased with the very interested and generally supportive response and thank the referees for their constructive criticisms. We carefully considered the reviewers' comments and addressed them as follows:

Reviewer #1:

In the manuscript "Ep400 deficiency in Schwann cells leads to persistent expression of early developmental regulators and peripheral neuropathy", Fröb et al. investigate the role of the chromatin remodeler Ep400 in Schwann cell development. The authors found that the conditional deletion of Ep400 in this lineage leads to peripheral neuropathy, by preventing the downregulation of transcription factors involved in early Schwann cell development. These findings are very interesting and are in line with recent findings that abnormal expression of early regulators in mature neural cells can have pathological consequences. The authors are very thorough in their analysis and I deem that the paper should be accepted to publication in Nature Communications, pending that the authors address the following points:

Response: We thank the referee for the encouraging and supportive assessment.

The authors focus on the deposition of H2A.Z by EP400, but this chromatin remodeler has been also to be involved in the deposition of the histone variant H3.3, which is also involved in transcriptional activation (Pradhan et., Mol Cell 2016). Given that the authors also observe effects on H3K27ac, the role on H3.3 might be even more relevant. Additional experiments assessing the exact genomic location of Ep400 (by ChIP-Seq) in the same samples where H2A.Z and H3K27ac was analysed and the effects of Ep400 knockdown on H3.3 deposition would be of interest. Nevertheless, given the dramatic phenotype the authors observe upon cKO of Ep400 and that they already provide mechanistic insights for this phenotype, I consider that these experiments are not absolutely required for the current manuscript to be accepted

and can be investigated in follow up studies. They should nevertheless be discussed in the paper.

Response: We agree with the referee that additional ChIP-Seq studies for H3.3 and Ep400 would be interesting. We actually tried to perform such studies. However, two commercially available (“ChIP grade”) antibodies against H3.3 did not work in our hands. This may also explain why published ChIP-Seq studies on H3.3 in cellular systems mostly use tagged versions of the histone variant in combination with antibodies against the tag. As far as ChIP for Ep400 is concerned, our homemade antibody works in principle. However, we have not yet succeeded in obtaining the necessary sensitivity for high-quality ChIP-Seq studies. We are therefore unable to include H3.3 and Ep400 ChIP-Seq experiments in the revised version. However, we have discussed the issue as suggested by the referee on p.3, bottom and p18, first paragraph of the revised manuscript.

The authors observe a redistribution of the H3K27ac peaks (Figure 3d) and also of H2A.Z peak (Figure 3e) in the cKO:

o The white-to-green scales in the 2 panels of Figure 3d should be the same, so the graphs can be directly comparable; the same applied to the blue scales in Figure 3e

Response: Scales were adjusted as requested.

o There are a substantial number of genes that appear to gain H3K27ac in the cKO in Figure 3d. What are these genes? Have the authors performed gene ontology on these?

Response: By gene ontology, genes that gain H3K27ac in the cKo are mostly associated with immune and inflammatory response and thus correlate well to genes showing upregulation in the cKo by RNA-Seq. The corresponding data are added as new Suppl. Fig. 7a and are described on p.11, second paragraph of the revised manuscript.

o Also, there are a substantial number of genes that appear to gain H2A.Z, but these appear to maintain their H3K27ac status (Figure 3e, panels to the right), so they are most likely not the same genes observed in Figure 3d to gain H3K27ac. The authors mention that these genes are upregulated by RNA-Seq analysis, but does this happen through a H3K27ac independent mechanism? The authors should elaborate on this point.

Response: For the revised version, we specifically analyzed how H2A.Z and H3K27ac occupancy changes around the TSS of genes that are upregulated in the cKO. These studies

showed that 37% of the upregulated genes with H2A.Z gain also acquired H3K27ac. 52% gained H2A.Z but were already H3K27ac-positive before. The remaining 11% gained H2A.Z, but remained H3K27ac-negative. Whether this is indicative of a small fraction of active promoters without H3K27ac enrichment or whether there is a simpler technical explanation is currently unclear. This is described on p.12, second paragraph of the revised manuscript.

- The authors have focused their ChIP-Seq analysis at TSSs, and thus promoters. However, H3K27ac also associates with enhancers, and H2A.Z as also been shown to be depleted at enhancers upon EP400 knockdown. The authors should also perform an analysis at the enhancer level.

Response: We like to mention that most of the analysis performed on H2A.Z was not restricted to promoter regions, but genome-wide so that it included enhancer regions as well. To specifically analyze H2A.Z at the enhancer level, we performed additional analyses in which we focused selectively on those enhancers that are normally active in early postnatal Schwann cells according to the published literature (Lopez-Anido et al., 2015). Interestingly, we found that the vast majority of these enhancers retained their H2A.Z status, as did the corresponding promoter regions. This confirms other data in the manuscript as it shows that H2A.Z occupancy is not so much altered on the regulatory regions of those genes that are normally expressed in myelinating Schwann cells, but rather affects genes that are not expressed in myelinating Schwann cells. The novel analysis on Schwann cell enhancers and promoters is included as novel Fig. 3f,g and is described and discussed on p.12, third paragraph of the revised manuscript.

In page 10, the authors mention “Arrangement of genes in the same order after ChIP-Seq of chromatin from Ep400 PNS nerves revealed a substantial redistribution of the promoter-associated H3K27ac modifications (Fig. 3d, right) indicating that RNA-Seq likely underestimates the occurring changes in transcription at the cellular level, because RNA-Seq determines transcript levels as an average over all cell types in the nerve.” The ChIP-Seq was performed also at a bulk level, so it would entail similar challenges, so I recommend the authors to reformulate the last part of the sentence.

Response: We agree with the referee and have removed the sentence from the manuscript.

Do the 12 transcription factors up-regulated in the RNA-Seq and increased H2A.Z peaks exhibit higher levels of H3K27ac? Since the authors focus on this histone modification early in the manuscript, they should comment on it.

Response: They all do. This is now mentioned on p.14, top of the revised manuscript and shown for Tfp2a and Pax3 in Fig. 4f,g and Suppl. Fig. 7d,k. In case that the referee thinks it is necessary to explicitly show the data for all 12 factors, we would be more than happy to add the corresponding IGV tracks on request.

In figure 1r and v, the authors state that statistical significance was determined by two-tailed Student's t-test, which is not the correct test to be used, since the data appears to be normalized to ctrl, which then does not present any variance. This would explain why the observed statistical significance for a 2.85% difference in figure 1v, which is unlikely to have biological relevance. The same applies to figure 2l, where an ANOVA analysis without normalization would be more advisable. The authors should also state the number of ns in all the figures.

Response: At the stages analyzed in Fig. 1r,v, all large caliber axons are in fact myelinated in controls so that no normalization was carried out. In Fig. 2l, comparison was always for a particular transcript and age between the control and the mutant. Therefore, the Student's t-test appears appropriate in both cases. To avoid misunderstandings, we provide a better explanation in the corresponding figure legends of the revised version (p.32, bottom and p.33, bottom).

In Figure 2e-k, the authors observed an increase in Oct6+ promyelinating Schwann cells, but then observe a dramatic decrease in Krox20+ myelinating cells. They also observe increased proliferation and cell death and an increase in microglia. While the conclusions of the authors that "Schwann cell differentiation in Ep400 PNS mice is disturbed early during myelination. This is accompanied with a slightly increased rate of apoptosis that is compensated by a similarly increased rate of proliferation." are plausible, these conclusions cannot be directly taken from the data, since the authors present quantifications for proliferation and apoptosis in general without correlating with the cell type. As such, the authors should show countings of Ki67+/Sox10+ cells, cCasp3+/Sox10+ and Ki67+/Iba1+ cells, cCasp3+/Iba+.

Response: We agree with the referee and have recounted the number of Ki67+ and cCasp3+ Schwann cells. This did not change the overall conclusions. The new data were included as Fig. 2h,i in the revised manuscript and the old quantifications were moved to Suppl. Fig. 5f,h. We could not perform analogous countings for Ki67+ and cCasp3+ macrophages because the Iba1 antibody has been generated in the same species as the Ki67 and cCasp3 antibodies. However, from comparison of the total number of proliferating cells and the number of

proliferating Schwann cells it is clear that the nerve contains additional proliferating cells. These are very likely macrophages. This is discussed on p. 9, top of the revised manuscript.

- The authors should add error bars in Figure 2b

Response: Error bars were added as requested (now Fig. 2a).

- The authors perform RNA-Seq in the sciatic nerve as bulk, which lead to a significant immune signature that could have masked the effects on Schwann cells. Ideally the authors should have done instead RNA-Seq on Sox10+ Schwann cells. While this would have significantly improved the paper, the authors did a very thorough and persistent analysis of their data, which allow them to identify the changes in early regulators, which they elegantly show later in the paper that are contributors to the observed phenotype.

Response: We were unable to acutely isolate sufficient numbers of mouse Schwann cells from dissociated nerve tissue by FACS or other methods. The only other option would have been to amplify Schwann cells in culture and place them in differentiating conditions before RNA-seq. However, even under differentiating conditions, mouse Schwann cells do not myelinate. It is not clear how this would have affected expression patterns and whether RNA-Seq on isolated Schwann cells would have been more informative. We therefore appreciate that the referee does not insist on a repetition of the RNA-Seq experiments on purified Schwann cells.

In Figure 4g, the authors show a graph with the quantification of % Sox2/Sox10, but which is very informative, but this is not mentioned in the Figure legend, where only the double staining Sox2 and Krox20 is mentioned.

Response: The quantification in Fig. 4g is indeed a quantification of the fraction of Krox20-expressing Schwann cells that are also labelled by Sox2. The y axis was accidentally mislabeled. We apologize for this mistake and have corrected it in the revised version (now Fig. 4h).

The authors mention that “these dko mice (Tfap2a and Ep400) were less affected than Ep400 PNS mice in their ability to move in the cage and the majority was still alive at five months of age.” A survival curve as in Supplementary Figure 2 would be advisable.

Response: A survival curve was added as Suppl. Fig. 8f as suggested by the referee.

- *What statistical test was done for Figure 5b? It is not possible to have statistical significance in Oct6 between EP400delta PNS and dko, the difference is minimal and the variation between the individual points too large*

Response: Thank you for noting. We very much agree. The bar had accidentally shifted to a wrong position during figure preparation. The statistically significant difference is between the control and the dko. This was corrected in the revised version (now Fig. 5b-d). ANOVA was used for statistics.

- *In Figure 5l, ANOVA should be used for statistical testing*

Response: We agree and have changed statistics accordingly (now Fig. 5n).

Reviewer #2 (Remarks to the Author):

In this manuscript, Fröb et al. studied the functions of the chromatin-remodeling complex Tip60/Ep400 in Schwann cell development. This complex allows the exchange of the histone H2A by the histone variant H2A.Z in nucleosomes. Fröb et al. employed mouse genetics to specifically ablate in Schwann cell precursors the ATPase subunit Ep400, without which the remodeling complex is not active. By using this approach, the authors show an important function of this chromatin-remodeling complex in postnatal myelination. In addition, they identify the transcription factor Tfap2a as a major target of the Tip60/Ep400 complex by showing that it is strongly upregulated in Ep400 KO Schwann cells and that the ablation of Tfap2a in Schwann cells in addition to Ep400 rescues partially the myelination defect of Ep400 KO Schwann cells. Overall this study is interesting, showing important functions of the chromatin-remodeling complex Tip60/Ep400 in postnatal myelination by the regulation of a set of specific target genes. However, I have a few comments below to strengthen the study.

We thank the referee for the very positive overall assessment.

1/ Figure 1d: What is the identity of Sox10-negative cells expressing Ep400 in Ep400ΔPNS sciatic nerves? These cells seem much more abundant in KO nerves. Are they macrophages? Please, add a double staining of Ep400 with a macrophage marker and DAPI labeling.

Response: To address the issue, we performed additional immunohistochemical stainings. We found that Ep400 is present in all nuclei of spinal and sciatic nerves. Co-immunohistochemistry furthermore identified the Sox10- Ep400+ cells in the nerve as either macrophages, T cells, perivascular smooth muscle cells, pericytes, endothelial cells or fibroblasts. These data were added as novel Suppl. Fig. 1k and are discussed on p.5, third paragraph of the revised manuscript.

Page 5: “Ep400 Δ NC mice exhibit multiple neural crest-related abnormalities...”

Please, either explain what type of neural crest-related abnormalities are observed and document these observations or remove the part on neural crest cells.

Response: We observed a series of severe craniofacial malformations including orofacial clefting that are sufficient to explain the perinatal death. Data are added as novel Suppl. Fig. 2f and discussed on p.6, top of the revised manuscript.

3/ Is it really necessary to write in the figure legends all numerical values? It renders legends very difficult to read. Why not instead adding an excel file in supplementary data to show all these values, if necessary?

Response: We followed the referee’s suggestion and moved the numerical values from the legends to Supplementary Tables.

4/ Fig. 2b: Has the relative distribution of internodal length been calculated only for 1 control and 1 Ep400 Δ PNS? It should be done on sciatic nerves of at least 3 animals for each group. Please, add the required number of animals for this analysis, and add error bars and statistics.

Response: Internodal length has indeed been calculated from 3 animals for each group. Error bars and statistics were forgotten in the original version and are now added in the revised version (now Fig. 2a).

5/ Fig. 2c-d: It would be more accurate to measure nodal and paranodal lengths by electron microscopy rather than by immunofluorescence of nodal and paranodal components. An average of 1 μ m for the nodal length found in KO is a value usually found in control animals. Measuring these parameters on longitudinal electron microscopy images would lead to more accurate values and strengthen this point.

Response: The original quantification of nodal and paranodal lengths were performed by standard microscopy. When repeated on a confocal microscope, the absolute values changed and conformed to those in the literature. We included the confocal measurements as figure into the revised manuscript (see novel Fig. 2b-d). We also looked at node and paranode in longitudinal sections by EM. The obtained values were in the same range as those obtained by confocal microscopy, but relatively few for statistics. EM data are mentioned in the text on p. 8, top and exemplarily shown as Suppl. Fig. 3o-r.

Fig. 2h-k: representative photos for KI67, TUNEL, cCasp3 and Iba1 should be added in supplementary figure, such as it is already done in Suppl. Fig.3 for Sox10, Oct6 and Krox20.

Response: As suggested we included representative photos for KI67, TUNEL, cCasp3 and Iba1 as Suppl. Fig. 5e,g,i,j in the revised manuscript.

Fig. 4g: I do not understand the bar graph under the co-staining Sox2/Krox20... The labeling seems wrong. In addition, values on the y axis cannot be a ratio and do not seem either to be a percentage. Please, correct.

Response: Thank you for noting. Labelling was indeed wrong. Shown was the fraction of Krox20-positive Schwann cells with additional Sox2 expression. In the revised version of the manuscript, the label for the y axis was corrected in (now) Fig. 4h.

In dko nerves, myelination is improved as compared to Ep400ΔPNS. What about nodal and paranodal lengths, proliferation and apoptosis? In addition, the g ratio for Tfap2aΔPNS does not appear on the figure. Please, add these data.

Response: As requested by the referee, we added more data to phenotypically characterize the dko nerves. This included proliferation, apoptosis, macrophage infiltration, the amount of myelin debris, the number of myelin outfoldings and Remak bundles, nodal and paranodal lengths. All parameters except nodal and paranodal length as well as the number of Remak bundles exhibited statistically significant improvements relative to Ep400 Δ PNS nerves. For many of these parameters, analysis was expanded from P21 to P0, P9 and 2 months. The corresponding quantifications are presented as novel Fig. 5a-d,o-u and Suppl. Fig. 8g,h and described and discussed on p.16 bottom and p.17 of the revised manuscript.

Reviewer #3 (Remarks to the Author):

This is a potentially interesting manuscript on the role of Tip60/Ep400 epigenetic modulator in Schwann cell development, which includes the phenotypic analysis of mice with genetic ablation using the Dhh Cre line and the molecular analysis of chromatin (isolated from sciatic nerves and immunoprecipitated with antibodies specific for the histone variant H2Az and for H3K27ac histone)

The proposed model is that in the absence of Ep400, H2Az persists on chromatin (as broader and higher peaks), thereby leading to inappropriate repression of transcription factors impact differentiation of Schwann cells and the authors support it by providing evidence of partial rescue by ablating also one of these factors (Tfap2a). While the phenotypic analysis and the molecular characterization individually reveal some interesting findings, it is unclear how the two are related. This is a serious concern, as the authors declare that the rationale for the molecular analysis is the identification of the "cause of the phenotype". Unfortunately, there is little attempt to connect molecular results to phenotype, with the exception of focusing on the 12 transcription factors, identified at the intersection of the "broad ChIP peaks in mutants" with the "upregulated transcripts".

Response: We thank the referee for the comments. Although we do not agree with the referee's statement that molecular results and phenotype are not linked, we understand that there may be differences in the assessment of how much data are needed to present a reasonable case for the linkage. To convince the referee, we have rewritten parts of the manuscript and added further analyses and data as suggested by the referee.

We now show in Fig. 3f,g, and Suppl. Fig. 8a-d that the regulatory regions of genes that are normally expressed in myelinating Schwann cells at early postnatal times overwhelmingly retain their H2A.Z status. Instead, we detected alterations in H2A.Z occupancy in the vicinity of several transcription factors that are normally expressed in immature cells and not in myelinating Schwann cells. These H2A.Z alterations were accompanied by increased H3K27ac and expression levels. At least for Tfap2a and Pax3 these alterations were stably maintained throughout the first two months as shown in novel Suppl. Fig. 7d-q. In our opinion, this is sufficient evidence for a link between Ep400 deletion and permanent upregulation of the transcription factors.

In the revised version, we have added further data as Fig. 4k-m (described on p.15, second and third paragraph) that document the ability of the identified transcription factors to interfere with Sox10-dependent expression of Krox20 and/or Krox20-dependent expression of several peripheral myelin genes. This is in line with published data and offers an explanation for the peripheral hypomyelination observed in the mutant. This explanation is furthermore supported

by the partial rescue of the Ep400 mutant phenotype by additional deletion of Tfp2a. This has been much better documented for the revised version than before and led to inclusion of new data in Fig.5a-u and Suppl. Fig. 8g,h.

In summary, we believe that we have build a convincing case for the proposed model and thereby connected molecular results and phenotypic defects in myelinating Schwann cells.

The phenotype appears to include multiple aspects of SC function and needs to be better characterized. there is not only hypomyelination at day P21 (as shown in Fig.1k and quantified in Fig 1q), but also myelin outfoldings(evident in Fig 1o and in Fig, 1x) and aberrant remak bundles. As presented the data are confusing and qualitatively presented. A thorough quantification of each finding (i.e. percentage of myelinated axons, number of remak bundles, axons with myelin outfoldings, axons with myelin degeneration) should be included for earlier (P7, P14, P21) and later (2mo, 6 mo).

Response: As requested by the referee, we added further quantitative data to characterize the phenotype in Ep400 mutant mice. This included quantifications of myelin outfoldings, axons with myelin degeneration, percentage of myelinated axons and number of Remak bundles. We performed these analyses until 2 months. The 6 months time point was not feasible within the limited time of a revision period. The new data are included as novel Suppl Fig. 3s-x in the revised version and are described and discussed on p. 7 of the revised manuscript.

The findings on the “strong alterations” of the node of Ranvier should be better presented and explained. The inclusion of sagittal ultrastructural images may provide insights. It is unclear whether the molecular analysis provides candidates that are able to explain these findings.

Response: As suggested, we included sagittal ultrastructural images in the revised version as novel Suppl. Fig. 3o-r. These images confirmed the altered length of node and paranode but did not reveal any other consistent structural abnormalities. In accord with RNA-Seq data, qRT-PCR experiments furthermore proved a reduction of paranodal proteins such as Neurofascin and Ankyrin-2. These data were also included in the revised version as novel Fig. 2m. While our data may not fully explain the changes at the node, they are in line with other defects of the myelin sheaths and do not represent the main or sole focus of our study.

The choice of the time points for RNA Seq and Chip Seq is questionable. The authors describe a reduced number of Krox20 cells already at birth, and yet the analysis is conducted much later, thereby reflecting not only changes in gene expression that may be directly related to gene ablation (and therefore “causal”) but also many transcriptional changes that are the

result of the cellular changes caused by gene ablation (and therefore “consequential”). Earlier analysis of transcripts may provide a better sense of what is causal. An example is the GO “immune response”, this is clearly a reactive response, as pointed out by the authors, and it does not shed light on the causal aspect of the phenotype. It is also unclear whether Ep400 regulates different genes at different times of development and that the phenotypes described at p21 is the results of a cumulative number of defects from distinct stages of SC development. At the very minimum, it would be important to mine the RNA Seq data for stage-specific genes and validate differential transcripts at multiple time points.

Response: We agree with the referee in the assessment that P9 may be already a bit late for RNA-Seq and ChIP-Seq analysis as there is already a reactive response. As evident from Fig. 2k, the reactive response was not present at birth. Thus, we may have obtained better results at P0. However, it would have been difficult to obtain sufficient amounts of high-quality RNA and chromatin at this earlier time point, and existing changes in Schwann cells may have been too subtle at P0 to be detectable in sciatic nerve tissue. We agree that the exact conditions of our RNA-Seq and ChIP-Seq studies have made the subsequent analyses more difficult. Nevertheless, these analyses were successful and allowed us to postulate a mechanism for which we provide substantial evidence.

Formally, we cannot rule out that Ep400 regulates different genes at different time points. However, we clearly show in the revised version for at least some of the identified transcription factors that they remain aberrantly expressed throughout the first two months of life (Fig. 4i) and that alterations in H2A.Z and H3K27ac occupancy are stable throughout the first two months (Suppl. Fig. 7d-q).

We have also mined the RNA-Seq data for stage specific genes and have analyzed their expression by qRT-PCR at multiple time points. These studies showed that expression of several genes that are characteristic of earlier stages of Schwann cell development remained fairly stable and normal in Ep400-deficient Schwann cells arguing that there is no global deregulation of gene expression in Schwann cells. These data are presented as novel Fig. 2l and p.9, bottom and p.11, first paragraph.

4. If there is already a problem at birth, why did the authors omit the P0 time point from panels 2h to 2k? Wouldn't be important to understand what are the first changes detected in these mice?

Response: As suggested by the referee, we included the P0 time point in Fig. 2h-k. The lower number of Krox20-positive cells is in fact the only change that we detected at P0.

5. The description of the RNA Seq data on page 9 needs to be substantially improved to include the total number of differential transcripts and a brief discussion of the analysis

Response: We added the total number of differential transcripts to the revised version of the manuscript as well as a brief discussion of the analysis (see p.10 and Materials and Methods section of the revised version).

6. the selection of the Gorilla software for GO analysis is questionable, especially since the authors continue to reference it throughout the paper. The authors may get better highlights from a GSEA analysis of the RNA Seq and selective relevant datasets. Similarly, for the ChipSeq analysis, the selected categories fail to shed light on the phenotype and I wonder whether a more in depth analysis, comparing multiple programs, including DAVID, may provide more meaningful GO categories.

Response: As suggested by the referee, we additionally performed GO analysis using DAVID and added GSEA analysis. The use of different software for the GO analysis did not change the results substantially and did not provide other GO categories. This is mentioned on p. 27, top of the revised manuscript. However, GSEA analysis was helpful in confirming downregulation of lipid metabolic genes. It also helped to unmask a general downregulation of myelination genes in the Ep400 mutant. The new data are included as Suppl. Fig. 6c-e and described and discussed on p.11, top paragraph of the revised manuscript.

The description of the ChipSeq data on page 11 needs to be reorganized and presented in a clearer manner. IGV curves need to be clearly presented in Fig 4 at multiple time points and, most importantly, to address the issue of causality, the ChIP data need to be validated in chromatin samples isolated from nerves at multiple time points (p9, p21 and 2 mo) and immunoprecipitated with antibodies for H2AZ and H3K27Ac from controls and mutants. Chip PCR and IGV curve at multiple time points are truly critical in Fig. 4 to support the main hypothesis of the paper, which in the absence of ep400, peaks of H2Az, which would be lost in the wild type mice, remain in the mutant.

Response: As requested by the referee, we validated our ChIP-Seq data for Pax3 and Tfp2a on additional chromatin samples at multiple time points after immunoprecipitation with H2A.Z and H3K27ac in ChIP-PCR. The obtained results are presented in novel Suppl. Fig. 7d-q and described on p.14, top of the revised manuscript. The ChIP-PCRs showed that the changes of H2A.Z and H3K27ac occupancy that we originally detected at P9 at these loci were stable in

the Ep400 mutant during the first 2 months. They thus support our previous data and assumptions.

8. While the data on the “partial rescue” are of potential interest, they are truly incomplete. A time-course of proliferation, apoptosis and ultrastructural characterization should be included. Without these data is unclear what the actual “rescue” is due to. There are clearly multiple aspects of SC biology that are impacted by Ep400 and the authors have the opportunity to shed some light on this important question.

Response: As requested by the referee, we expanded the quantitative analysis of the double knockout by including extra time points and by analyzing multiple parameters such as various Schwann cell markers, myelin sheath thickness, myelin outfoldings, degenerated myelin, nodal and paranodal lengths, proliferation, apoptosis and number of macrophages. These results are presented as several expanded and novel panels in Fig. 5a-u and Suppl. Fig. 8g,h in the revised manuscript. As described on p.16, bottom and p.17, top they show a partial rescue in nearly all parameters except nodal/paranodal length and number of Remak bundles.

9. There is an important inconsistency between the description of the phenotype of the Ep400 mutants, where “mostly large caliber axons appear to be affected” and the rescue in the triple mutant lacking also Tfap2a. It would be important to clarify.

Response: The inconsistency was not in the data, but in our original statement. The data show that myelin thickness is most strongly affected for large caliber axons in Ep400 mutants. In the double mutant myelin thickness is partially rescued for axons of all caliber sizes. This leads to a loss of statistically significant differences in the g-ratio of small caliber axons as compared to controls, whereas the large caliber axons still show increased g ratios. We apologize for our original statement and have deleted it in the revised version.

The authors should explain the rationale for focusing on this molecule. Why was it selected? is there a translational relevance due to mutations of the gene in peripheral neuropathies?

Response: We provide the rationale on p.15, fourth paragraph of the revised manuscript. Feasibility was an important aspect. Tfap2a expression is normally extinguished in Schwann cells at the time when Dhh expression starts. Therefore, we could be fairly sure that Dhh::Cre-dependent deletion of Tfap2a would not impede Schwann cell development on its own.

There is no known link of Tfp2a to peripheral neuropathies. If existent, such mutations would have to be gain-of-function mutations. So far only loss-of-function mutations have been identified as cause of Branchiooculofacial Syndrome.

Data obtained using the Sox10 cre line are briefly mentioned and not shown, due to the extensive repercussions on neural crest cell development. They may be included.

Response: Data on neural-crest related craniofacial malformations and orofacial clefting were included as novel Suppl. Fig. 2f and are described on p.6, top of the revised manuscript.

The definition of “bigger” peaks is a bit imprecise and would be best if replaced with the same terms used in the methods” “broader and higher” . The same applies to “smaller” peaks (they are “narrower”)

Response: We followed the referee’s suggestion.

The text of the manuscript contains a large amount of grammatical errors with present and past tenses incorrectly used (i.e. results should be all in past tense), colloquialisms (“if anything there was...” “goes along with ...”), unusual choice of terms (i.e. “disturbed” differentiation, “higher” age, “almost normalized”...), repetitions (“normally”... “normally” within the same sentence) and this is truly distracting for the reader. The text needs to be extensively revised.

Response: We apologize for our shortcomings as non-native speakers in the use of the English language. We made every attempt to correct the mistakes and had our manuscript proof-read by a number of colleagues. If we missed a few errors, we ask the referee to bear with us.

REVIEWERS' COMMENTS:

Reviewer #1 (Remarks to the Author):

The authors have answered appropriately to my previous comments and therefore I consider that the manuscript should be accepted for publication in Nature Communications.

Reviewer #2 (Remarks to the Author):

The authors have appropriately addressed my comments and the comments of the other reviewers. The manuscript is now improved and suitable for publication in Nature Communications.

Reviewer #3 (Remarks to the Author):

The Authors have been very responsive to the Reviewers ' suggestions and accurately and thoroughly addressed their concerns. The text revisions and the inclusions of new data have substantially improved the readability of the manuscript and rendered the story quite compelling

FRIEDRICH-ALEXANDER
UNIVERSITÄT
ERLANGEN-NÜRNBERG
MEDIZINISCHE FAKULTÄT

Emil-Fischer-Zentrum
Institut für Biochemie

Institut für Biochemie Fahrstrasse 17 91054 Erlangen

Lehrstuhl für Biochemie & Pathobiochemie
Prof. Michael Wegner

Telefon: (09131) 85-24638
Telefax: (09131) 85-22484
E-Mail: michael.wegner@fau.de

April 23rd, 2019

Point by point response to the reviewers' comments
on manuscript NCOMMS-18-30213A
by Fröb et al.

"Ep400 deficiency in Schwann cells causes persistent expression of early developmental regulators and peripheral neuropathy"

We were very pleased that all three referees acknowledged our efforts during the revision process and unanimously confirmed that we accurately and thoroughly addressed the referees' concerns and appropriately revised our manuscript. All three referees found the manuscript substantially improved and compelling. The referees had no further comments and recommended acceptance.

We like to thank the referees for their expert opinions, critical input and constructive criticism which has helped to strengthen the manuscript considerably.

Sincerely.

Michael Wegner

Direktor
Prof. Michael Wegner

Institutsanschrift
Fahrstrasse 17
91054 Erlangen

Telefon
09131 85-24620
Telefax
09131 85-22484

E-Mail
michael.wegner@fau.de
Internet
www.biochem.uni-erlangen.de